# Adaptive Physics Transformer with Fused Global-Local Attention for Subsurface Energy Systems

**Xin Ju** [1 2]  **Nok Hei (Hadrian) Fung** [3]  **Yuyan Zhang** [3]  **Carl Jacquemyn** [3]  **Matthew Jackson** [3]
**Randolph Settgast** [4 2]  **Sally M. Benson** [1 2]  **Gege Wen** [3 2]

## Abstract

The Earth's subsurface is a cornerstone of modern society, providing essential energy resources like hydrocarbons, geothermal, and minerals while serving as the primary reservoir for $CO_2$ sequestration. However, full physics numerical simulations of these systems are notoriously computationally expensive due to geological heterogeneity, high resolution requirements, and the tight coupling of physical processes with distinct propagation time scales. Here we propose the **Adaptive Physics Transformer** (APT), a geometry-, mesh-, and physics-agnostic neural operator that explicitly addresses these challenges. APT fuses a graph-based encoder to extract high-resolution local heterogeneous features with a global attention mechanism to resolve long-range physical impacts. Our results demonstrate that APT outperforms state-of-the-art architectures in subsurface tasks across both regular and irregular grids with robust super-resolution capabilities. Notably, APT is the first architecture that learns directly from HR-adaptive mesh refinement simulations. We also demonstrate APT's favorable scaling behavior and cross-dataset learning capability, positioning it as a robust and scalable backbone for large-scale subsurface foundation model development.

## 1. Introduction

Full physics numerical simulations of flow and transport through heterogeneous geological formations are a fundamental component for decision-making in subsurface energy processes (Bear & Cheng, 2010), such as geological carbon storage (GCS) (Strandli et al., 2014; Tang et al., 2021), geothermal resource development (Settgast et al., 2017), lithium brine extraction, and petroleum engineering (Aziz, 1979; Christie & Blunt, 2001). However, HF simulation of realistic subsurface projects presents significant computational challenges. The underlying physics involves complex coupled partial differential equations (PDEs) at different spatial and time scale, including multiphase flow dynamics, mutual component solubility, phase transitions, capillary effects, viscous fingering instabilities, gravitational override, and rock matrix interactions (Orr et al., 2007). Each is formulated as high-dimensional, nonlinear Partial Differential Equations (PDEs). In addition, subsurface geological formations have highly heterogeneous permeability and porosity that significantly influencing the pressure propagation and fluid flow patterns (Pini et al., 2012).

Data driven machine learning (ML) approaches provide a computationally efficient yet accurate alternatives to traditional numerical solvers for subsurface energy systems. Despite significant progress, current ML architectures face fundamental limitations in geometric flexibility and scalability (see detailed related works in Appendix A.). Convolutional Neural Networks (CNNs), Fourier Neural Operators (FNOs), and vision transformer-based (Mao et al., 2025) architecture are inherently restricted to structured, gridded discretizations (Wen et al., 2022; 2021), making them unsuitable for the unstructured meshes required to resolve complex geological features like faults or stratigraphic pinch-outs. Hierarchical frameworks such as Nested FNO (Wen et al., 2023b) and Nested Fourier-DeepONet (Lee et al., 2024) attempts to learn large-scale systems with semi-adaptive local grid refinement. However, they require training multiple separate models for a single system, which resulting in computationally intensive training pipelines that cannot generalize across varying semi-adaptive meshes. Graph-based simulators like MeshGraphNet (Pfaff et al., 2021) (MGN) excel at representing irregular geometries but are limited by the local nature of their message-passing kernels, which struggle to capture the long-range global dependencies essential for pressure propagation. Moreover, existing

[1]Department of Energy Science and Engineering, Stanford University [2]EarthFlow AI, Inc. [3]Department of Earth Sciences and Engineering, Imperial College London [4]Lawrence Livermore National Laboratory. Correspondence to: Xin Ju <ju1@stanford.edu, isaacju@earthflow.ai>.

*Proceedings of the 43[rd] International Conference on Machine Learning*, Seoul, South Korea. PMLR 306, 2026. Copyright 2026 by the author(s).

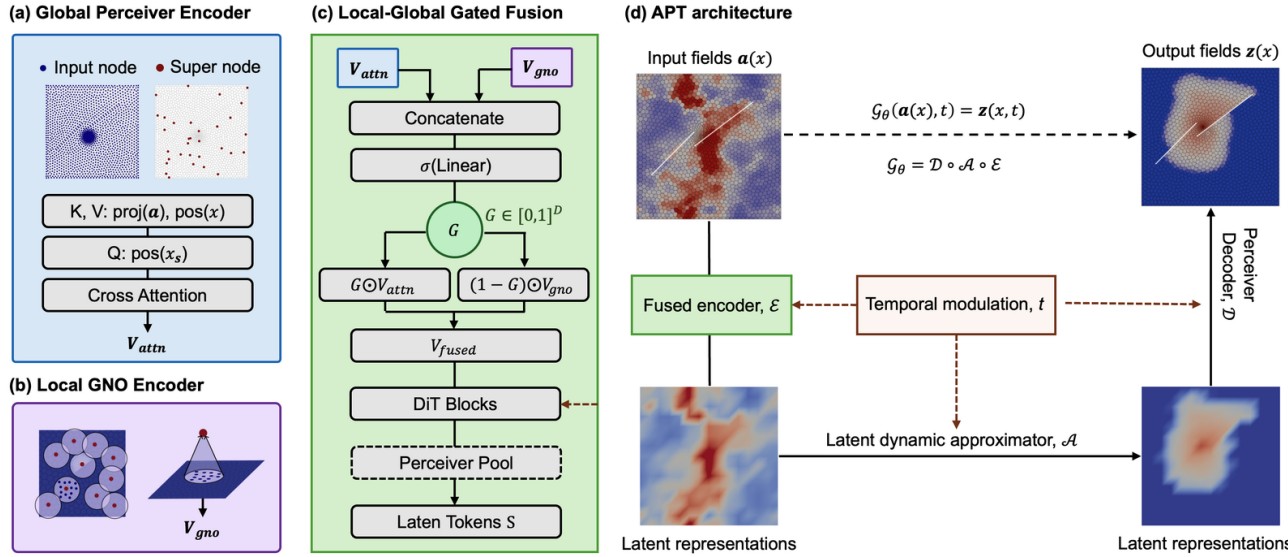

*Figure 1.* Architectural Overview. The fused encoder combines a **(a)** Global Perceiver Encoder that projects input features onto supernode queries via cross-attention with a **(b)** Local GNO Encoder that aggregates neighborhood information through radius graph pooling. A **(c)** Gated Fusion Mechanism adaptively combines global ($\mathbf{v}_{attn}$) and local ($\mathbf{v}_{gno}$) representations via a learned gate $G \in [0, 1]^d$, followed by DiT blocks and Perceiver pooling to produce fixed-size latent tokens. The overall pipeline **(d)** APT architecture maps input fields $a(x)$ through the fused encoder $\mathcal{E}$, latent dynamics approximator $\mathcal{A}$ with temporal modulation, and Perceiver decoder $\mathcal{D}$ to generate output fields $z(x,t)$ at arbitrary query locations.

architectures lack the capability of learning directly from adaptive mesh simulation datasets. As a result, current ML approaches force each application to be treated as an isolated task, precluding the benefits of large-scale pre-training.

Motivated by the limitations of previous models, the physics transformer-based (Alkin et al., 2024; Wu et al., 2024a) architectures have emerged as a powerful paradigm that utilizes latent representations derived from flexible encoders coupled with attention mechanisms. However, existing approaches in this family overlook a fundamental characteristic of subsurface systems: *intrinsic spatial heterogeneity*. In standard fluid dynamics or solid mechanics problems, the computational domain is typically homogeneous. In these cases, while the boundary geometry influences the field, the underlying medium properties often remain constant throughout the space (Wu et al., 2024a; Elrefaie et al., 2024b; Alkin et al., 2024). Even in multi-solid systems, where different objects may possess distinct mechanical properties, existing approaches typically assume each individual object to be internally homogeneous (Tao et al., 2025).

For subsurface systems, properties such as permeability can vary across several orders of magnitude within a few meters. Existing architectures designed for homogeneous domains are often inefficient to capture these high-frequency local property variations, leading to smoothed prediction of fluid or thermal plumes (Wen et al., 2022). To resolve this issue, we propose the Adaptive Physics Transformer (APT), which

introduces a fused mechanism that adaptively utilize local and global information. The architecture concurrently captures local heterogeneity via graph neural operator encoders and long-range dependencies via global perceiver (Figure 1), resolving the disparate propagation scales inherent in subsurface physics. Our contribution highlights include:

- **Direct Learning from Adaptive Meshes**: To our best knowledge, APT is the first neural operator to learn directly from adaptive mesh datasets with dynamic mesh optimization (DMO) algorithms. DMO simulations can significantly reduce computational costs during data generation processes. Directly training with DMO avoids error from interpolation.

- **Superior performance and Super-Resolution**: APT outperforms SOTA neural operators on various benchmark subsurface tasks. We also showcased the superior super-resolution capability against SOTA models.

- **Cross-Dataset Training**: The APT architecture can serve as a scalable backbone for massive pre-training due to its ability to learn across *across* subsurface datasets with distinct mesh and distribution.

## 2. Methodology

### 2.1. Learning Task Set Up

Let $a(x)$ denote the input that specifies a particular instance of a subsurface system, such as rock permeabil-

ity and initial conditions. The corresponding PDE solution consists of time-dependent physical fields, such as gas saturation $S_g(x, t)$ and pressure buildup $\Delta P_g(x, t) = P_g(x, t) - P_g(x, 0)$. We collectively denote these output fields by $z(x, t)$. We define $\mathcal{X}$ as the space of inputs $a(x)$, and $\mathcal{Z}$ as the space of solution functions $z : D \times T \rightarrow \mathbb{R}^{d_z}$, where $D \subset \mathbb{R}^d$ is the spatial domain and $T = [0, T_{\max}]$ is the time interval of interest.

The spatial-temporal evolution of the system can be described by a solution operator $\mathcal{S}_t : \mathcal{X} \rightarrow \mathcal{Z}_t$, which maps the initial system definition $a$ to the solution $z(\cdot, t)$ at a specific time $t$. For time-dependent PDEs, we can define an extended solution operator $\mathcal{G}^\dagger$ (Li et al., 2021). This operator $\mathcal{G}^\dagger$ can map the system definition $a$ and a query time $t$ to the solution at that time, $z(\cdot, t) = \mathcal{G}^\dagger(a; t)$.

The learning task for the APT model is to learn a neural operator $\mathcal{G}_\theta$, parameterized by $\theta$, that approximates $\mathcal{G}^\dagger$. Given a set of input functions $\{a_j\}$ drawn from a probability measure $\mu$ on $\mathcal{X}$, the corresponding true solutions at any time $t$ are $z_j(\cdot, t) = \mathcal{G}^\dagger(a_j; t)$. The APT model $\mathcal{G}_\theta(a_j; t)$ aims to predict this solution. For numerical implementation, the input functions $a_j(x)$ are discretized on a spatial mesh, resulting in $a_j \in \mathbb{R}^{N_{\text{points}} \times d_a}$. The output functions $z_j(x, t)$ are represented at discrete spatial locations and sampled time instances, $z_j(x_i, t_k) \in \mathbb{R}^{d_z}$.

## 2.2. Adaptive Physics Transformer Formulation

We approximate $\mathcal{G}^\dagger$ with a three-stage mapping: $\mathcal{G}_\theta = \mathcal{D} \circ \mathcal{A} \circ \mathcal{E}$, as shown in Figure 1. The fused encoder $\mathcal{E} : \mathcal{X} \rightarrow \mathbb{R}^{d_h}$ maps the input function into a finite-dimensional latent vector. The latent dynamics approximator $\mathcal{A} : \mathbb{R}^{d_h} \rightarrow \mathbb{R}^{d_h}$ models the action of the solution operator in the latent space. The decoder $\mathcal{D} : \mathbb{R}^{d_h} \rightarrow \mathcal{Z}$ reconstructs the output function, usually as pointwise values on the target grid or mesh.

## 2.3. Input Representations and Fused Encoder

The goal of the encoder is to map an input field evaluated at $N_{\text{points}}$ to a fixed-size latent array $S$, in other words, compress features into the latent space. Input features $a(x_v)$ (e.g., permeability, porosity) are first projected into a hidden space of width $d_h$ through a learnable linear map $W_f \in \mathbb{R}^{d_h \times d_a}$. To incorporate spatial context, we compute a positional encoding (PE) $\mathbf{p}(\mathbf{x}_v)$ as detailed in Appendix B.1. The resulting node embeddings $\mathbf{h}_v^{(0)} = W_f a(x_v) + \mathbf{p}(\mathbf{x}_v)$ are further conditioned by a learnable time embedding $\mathbf{e}_{\text{cond},t}$. This initial temporal conditioning ensures the encoder is aware of the specific timestamp $t$ during the feature extraction phase.

**Fused Encoder and Parallel Local-Global Pathways.** A key architectural contribution of APT is the novel design of the parallel local-global feature extraction. Unlike previous operators that rely on either purely global feature extraction (e.g., FNO (Li et al., 2021), Transolver (Wu et al., 2024a)) or local graph-pooling encoder for feature extraction followed by self-attention layers to gain global perception (e.g., UPT (Alkin et al., 2024)), APT employs a fused encoder (Figure 1 (c)) designed to *simultaneously* capture multi-scale physics. This is especially important for subsurface systems where both sharp local interactions (e.g., saturation fronts, thermal fronts, and fault interfaces) and long-range pressure propagation must be resolved accurately.

Specifically, the Global Perceiver Branch (Figure 1(a)) uses cross-attention to project $N_{\text{points}}$ onto $N_s$ learnable supernode queries $\mathbf{Q} = \text{pos}(x_s)$, capturing long-range dependencies without the quadratic complexity of standard self-attention. Meanwhile, the Local GNO Branch (Figure 1(b)) captures fine-grained spatial heterogeneity using a Graph Neural Operator with radius graph construction and mean aggregation (Li et al., 2020; Hamilton et al., 2017).

**Gated Fusion and Latent Dynamics.** We integrate the global representations $\mathbf{v}_{\text{attn}}$ and the local representations $\mathbf{v}_{\text{gno}}$ via a Local-Global Gated Fusion mechanism (Figure 1(c)). This design is introduces complementary inductive biases: the global branch ($\mathbf{v}_{\text{attn}}$) mimics integral operator behaviors, while the local branch ($\mathbf{v}_{\text{gno}}$) adheres to the sparsity inherent in differential operators.

While gated fusion has been explored in prior hybrid neural operators (Wen et al., 2022; Liu & Tang, 2025; Liu-Schiaffini et al., 2024), they were often introduced as a decoding mechanism. Instead, APT framework uniquely employs these fused representations to *construct the latent token* space. To achieve this, we employ a learnable gating parameter $G \in [0, 1]^{d_h}$ to adaptively modulate the integration of features at each spatial location:

$$\mathbf{v}_{\text{fused}} = G \odot \mathbf{v}_{\text{attn}} + (1 - G) \odot \mathbf{v}_{\text{gno}}. \tag{1}$$

This mechanism enables APT to flexibly balance the contribution of global propagation trends against local conservation constraints. The resulting $\mathbf{v}_{\text{fused}}$ is subsequently projected into latent tokens $N_{\text{tokens}}$ and processed by a sequence of DiT blocks, subject to the temporal and conditional modulation detailed in Section B.1.

## 2.4. Approximator

The latent dynamics approximator $\mathcal{A}$ models the system's temporal evolution in latent space. It takes as input the fixed-size encoded state $\mathbf{z}_{\text{enc}} \in \mathbb{R}^{N_{\text{lat}} \times d_h}$ and a target time $t$ (via its embedding $\mathbf{e}_{\text{cond},t}$), and outputs the time-specific latent representation $\mathbf{z}_{\text{lat}}(t)$:

$$\mathbf{z}_{\text{lat}}(t) = \mathcal{A}(\mathbf{z}_{\text{enc}}; \mathbf{e}_{\text{cond},t}) \tag{2}$$

The approximator $\mathcal{A}$ consists of a small stack of Transformer layers that apply self-attention over the $N_{\text{lat}}$ latent tokens. Each latent vector in $\mathbf{z}_{\text{enc}}$ attends to all others to capture temporal and relational dependencies. The attention is computed using scaled dot-product self-attention:

$$\text{SelfAttend}(\mathbf{z}_{\text{enc}}) = \text{softmax}\left(\frac{QK^\top}{\sqrt{d_h}}\right) V \in \mathbb{R}^{N_{\text{lat}} \times d_h} \quad (3)$$

where the queries, keys, and values are computed via learned linear projections.

$$Q = \mathbf{z}_{\text{enc}} W^Q, \quad K = \mathbf{z}_{\text{enc}} W^K, \quad V = \mathbf{z}_{\text{enc}} W^V, \quad (4)$$

where $W^Q, W^K, W^V \in \mathbb{R}^{d_h \times d_h}$. The time embedding $\mathbf{e}_{\text{cond},t}$ is injected into the attention block (and subsequent feed-forward sublayers), ensuring that $\mathcal{A}$ produces temporally conditioned outputs.

Since the attention operates over a fixed number of latent tokens ($N_{\text{lat}}$), the computational complexity remains bounded at $\mathcal{O}(N_{\text{lat}}^2 \cdot d_h)$ regardless of the input size.

### 2.5. Decoder

The decoder $\mathcal{D}$ transforms the time-specific latent state $\mathbf{z}_{\text{lat}}(t) \in \mathbb{R}^{N_{\text{lat}} \times d_h}$ into physical output fields $\hat{z}(x, t)$ at user-specified spatial query points $x_{\text{query}}$ and time $t$. To achieve this, we employ a cross-attention operation. Each query location $x_{\text{query}}$ is embedded (e.g., via positional encodings) and combined with the time embedding $\mathbf{e}_t$ to form spatial-temporal query vectors $Q_{\text{out}} \in \mathbb{R}^{N_{\text{query}} \times d_h}$. These queries attend to the latent array $\mathbf{z}_{\text{lat}}(t)$, which serves as both keys and values:

$$\hat{z}(x_{\text{query}}, t) = \text{CrossAttend}\left(Q_{\text{out}}(x_{\text{query}}, \mathbf{e}_{\text{cond},t}), \mathbf{z}_{\text{lat}}(t), \mathbf{z}_{\text{lat}}(t)\right) \quad (5)$$

This set up allows querying at arbitrary spatial locations with computational complexity $\mathcal{O}(N_{\text{query}} \cdot N_{\text{lat}} \cdot d_h)$. A small linear layer is then applied to the attention output to produce the final physical predictions. All decoder operations are also conditioned by the time embedding $\mathbf{e}_{\text{cond},t}$. For subsurface applications with sharp fronts, we found empirically that stacking multiple cross-attention layers also improves the reconstruction of localized features.

### 2.6. Loss Functions and Training Strategy

The APT model is trained in a directly one-step manner from the initial condition to any query time step $t$. Unlike autoregressive physics transformers (UPT (Alkin et al., 2024), MGN-LSTM (Ju et al., 2024)) that require uniform temporal discretization for latent rollout, APT's direct temporal conditioning and training strategy allows native learning from simulations with temporal queries with irregular spacing. This is critical for subsurface systems where temporal resolution often varies with solution dynamics.

We use a relative $l_p$-loss to train the model. The training objective is to minimize the discrepancy between the model's predictions $\hat{z}(x, t)$ and the ground-truth solutions $z(x, t)$ over a set of $k$ randomly sampled spatiotemporal query points $\{(\mathbf{x}_i, t_i)\}_{i=1}^k$. The loss is defined as follows:

$$\mathcal{L} = \frac{\sum_{i=1}^k \|z(\mathbf{x}_i, t_i) - \hat{z}(\mathbf{x}_i, t_i)\|_p}{\sum_{i=1}^k \|z(\mathbf{x}_i, t_i)\|_p + \epsilon} \quad (6)$$

where $\epsilon$ is a small constant added for numerical stability.

## 3. Experiments

We evaluate APT on five subsurface energy benchmarks (Figure 2). This diversity tests APT's ability to generalize across mesh types (structured, unstructured, adaptive, nested), scales (thousands to millions of cells), and physics (single-phase, multiphase, compositional flow, and heat transfer).

### 3.1. Irregular Grid with Faults - $CO_2$ storage

Faulted system is a fundamental challenge for grid-based neural operators. Faults create discontinuities that require unstructured meshes for accurate modeling. We evaluate on a 2D faulted $CO_2$ storage benchmark with multiphase flow simulations from the GEOS numerical simulator (Settgast et al., 2024), where each case has a unique mesh adapted to varying fault and well configurations. Dataset details, baselines, and training configurations, training costs are provided in Appendix C.1.2. Table 1 compares APT against grid-based methods (U-Net, U-FNO) that require interpolation to regular grids, graph-based methods (MGN, MGN-LSTM) that operate directly on irregular meshes, and transformer-based and hierarchical neural operators-based architectures (UPT (Alkin et al., 2024), Transolver (Wu et al., 2024a), MINO (Shi et al., 2025), HOOD (Grigorev et al., 2023)).

Table 1. Results on 2D faulted $CO_2$ storage. APT with fused attention achieves the lowest error on both pressure and saturation.

| Model | $\delta P_g$ (%) | $\delta S_g$ (%) | Params |
|---|---|---|---|
| U-Net | 0.37±0.16 | 2.87 ± 1.08 | 1.36M |
| U-FNO | 0.34 ± 0.17 | 2.4 ± 1.10 | 1.48M |
| MGN | 1.00 ± 0.40 | 7.24 ± 1.30 | 1.94M |
| MGN-LSTM | 0.20 ± 0.15 | 1.20 ± 0.30 | 1.38M |
| *Transformer / hierarchical baselines* | | | |
| Transolver | 0.87 ± 0.76 | 3.82 ± 0.57 | 1.59M |
| MINO | 0.94 ± 0.75 | 6.89 ± 2.41 | 1.64M |
| UPT | 0.43 ± 0.03 | 4.33 ± 0.36 | 1.68M |
| HOOD | 0.79 ± 0.02 | 6.48 ± 0.94 | 1.98M |
| *APT variants (ablation)* | | | |
| APT (global only) | 0.48 ± 0.25 | 1.50 ± 0.60 | 1.36M |
| APT (local only) | 0.18 ± 0.09 | 0.36 ± 0.17 | 1.36M |
| **APT (fused)** | **0.11±0.07** | **0.32±0.16** | 1.38M |

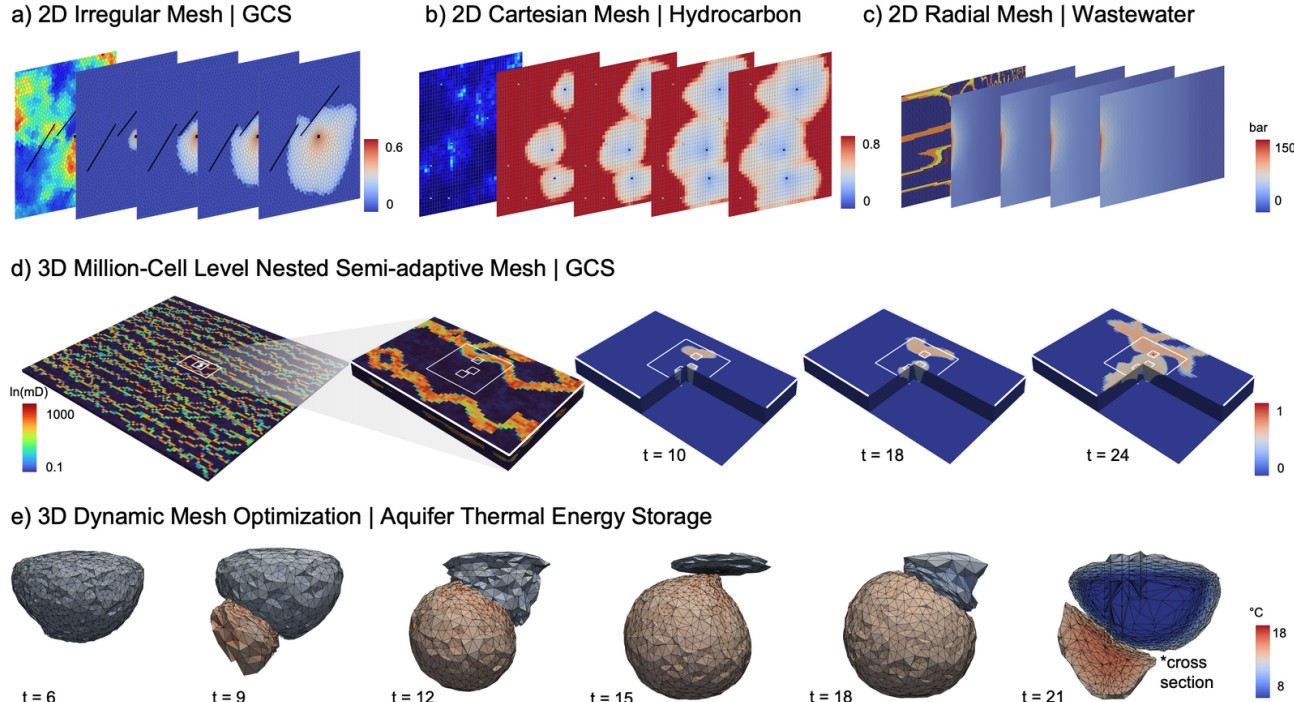

*Figure 2.* Benchmark datasets for evaluating APT across diverse subsurface applications. (a) to (d) each displays input parameter fields (e.g., permeability) alongside the temporal evolution of output fields (e.g., saturation, pressure, temperature). **(a)** 2D Geologic Carbon Storage (GCS) utilizing an irregular mesh to resolve complex fault geometries. **(b)** 2D Hydrocarbon extraction on a Cartesian mesh with varying well configurations. **(c)** 2D Wastewater injection system modeled on a radial mesh. **(d)** 3D Basin-scale GCS featuring a million-cell nested semi-adaptive mesh with local grid refinement (LGR), supporting both Gaussian and channelized permeability geomodels. **(e)** 3D Aquifer Thermal Energy Storage (ATES) employing dynamic mesh optimization to track moving thermal fronts.

APT achieves 0.11% pressure error and 0.32% saturation error, outperforming MGN-LSTM by $1.8\times$ on pressure and $3.75\times$ on saturation. The ablation (bottom rows) reveals the importance of fusing local and global attention: global-only attention struggles with the permeability heterogeneity at fault interfaces, while local-only message passing misses long-range pressure propagation. Fusing both mechanisms captures the interplay between local fault geometry and global pressure dynamics.

### 3.2. Cartesian Grid - Hydrocarbon Extraction

We evaluate on the public two-phase oil-water flow benchmark from Badawi & Gildin (2025), which uses a 40×40 Cartesian grid with heterogeneous Gaussian permeability fields. Here APT is trained to generalize across varying permeability fields, well configurations, and operating conditions. Notably, models are trained on cases with *only* 6 wells but tested on cases with 3–11 wells, requiring extrapolation to unseen well counts. Dataset details and governing equations are provided in Appendix C.1.1.

Table 2 presents the performance on this test set. APT demonstrates exceptional efficiency, reducing saturation error ($\delta S_w$) by over 50% compared to the strongest baseline

*Table 2.* Results on Cartesian grid hydrocarbon benchmark. APT achieves the best saturation prediction with $2.5\times$ fewer parameters than U-FNO.

| Model | $\delta P$ (%) | $\delta S_w$ (%) | Params |
|---|---|---|---|
| FNO | $1.85\pm1.07$ | $1.28\pm0.98$ | 31M |
| U-FNO | $\mathbf{0.57\pm0.41}$ | $0.66\pm0.53$ | 33M |
| **APT (ours)** | $0.60\pm0.53$ | $\mathbf{0.32\pm0.14}$ | **12M** |

(0.30% vs. 0.66% for U-FNO). Moreover, the superior fidelity of the saturation fronts of APT's is visually evident in Figure 3(b), where it captures sharp oil-water interfaces that the baseline methods tend to smooth out. Pressure predictions ($\delta P$) also remain comparable to the strong baseline model despite the $2.5\times$ smaller model size (12M vs. 33M).

### 3.3. Adaptive Mesh - Aquifer Thermal Energy Storage

We then evaluate APT on a challenging dataset generated by Dynamic Mesh Optimization (DMO) (Regnier et al., 2022) for Aquifer Thermal Energy Storage (ATES) (Jackson et al., 2024). ATES is a shallow geothermal technology that seasonally stores and retrieves thermal energy through

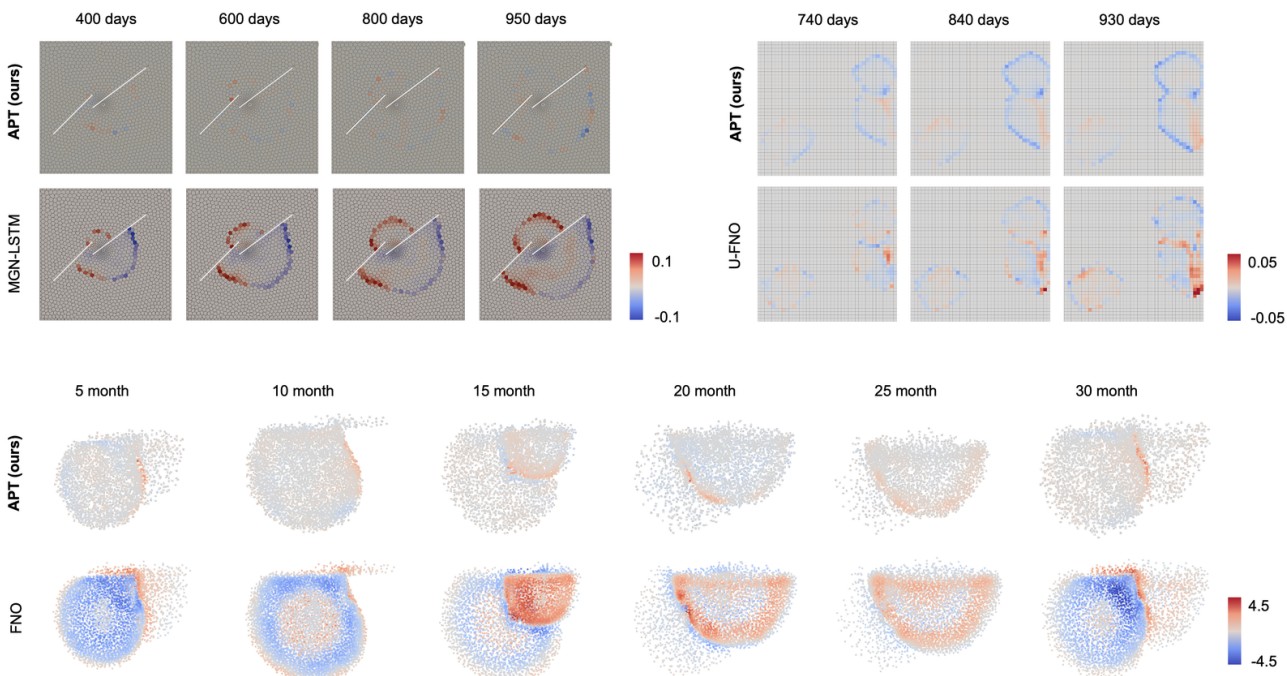

*Figure 3.* Visualization of prediction error and state evolution across different subsurface applications. Rows compare APT against baseline models (MGN-LSTM, U-FNO, and FNO) at multiple time steps ($t$). **(a)** Saturation error maps ($\delta S_g$) for the 2D irregular mesh geologic carbon storage dataset, showing APT's stability near complex fault geometries. **(b)** Water saturation error maps ($\delta S_w$) for the 2D Cartesian hydrocarbon dataset. **(c)** Spatiotemporal temperature evolution for the 3D dynamic mesh ATES dataset, where APT maintains higher fidelity to the thermal front compared to FNO. Errors and physical quantities are normalized within each sample for comparative visualization.

paired warm and cold wells to provide heating and cooling for buildings. The dataset comprises 840 scenarios simulated over 10-year seasonal injection-extraction cycles (240 timesteps) using numerical simulator IC-FERST. More details on the simulation setup are given in Appendix C.2.1.

As illustrated in Figure 2(e), DMO concentrates mesh resolution near the injection wells and the thermal fronts to maintain high fidelity. This results in meshes with *time-varying topology and node counts*—a fundamental challenge for standard neural operators that assume fixed spatial discretizations. Grid-based baselines (e.g., U-Net, FNO) are forced to interpolate these adaptive unstructured meshes onto a fixed regular grid, inevitably introducing smoothing artifacts and losing the precise geometric information preserved by DMO. Graph based methods, such as MGN (Pfaff et al., 2021) and its variants (Ju et al., 2024), also fails to make prediction for this system as they requires the same graph topology for each time step.

Here we demonstrate that APT learns continuous field mappings, allowing it to operate directly on the raw, time-varying DMO point clouds without interpolation. To handle the extreme sparsity of injection wells (<0.1% of domain volume), we employ anchor supernodes that deterministically preserve critical boundary regions. We further com-

pare APT against three recent mesh-native transformer baselines. UPT decouples input and output mesh topologies through learned supernode queries. Transolver applies slice-based attention to fixed unstructured meshes. MINO is a mesh-informed neural operator originally proposed for generative tasks on fixed irregular meshes. More details on the dataset preprocessing, as well as APT and other baseline implementations are given in Appendix C.2.1.

*Table 3.* Results on adaptive mesh ATES benchmark. APT learns directly from DMO without interpolation, achieving $4\times$ lower error than grid-based baselines.

| Model | Rel. $L_2$ | $R^2$ (%) | Params |
|---|---|---|---|
| U-Net (interpolated) | 0.084 | 69.9 | 77 M |
| FNO (interpolated) | 0.046 | 90.6 | 21 M |
| *Mesh-native baselines* | | | |
| Transolver | 0.0237 | 98.6 | 17.5 M |
| UPT | 0.0401 | 96.0 | 19.6 M |
| MINO | OOM[a] | | – |
| *APT variants* | | | |
| APT (w/o fused) | 0.014±0.01 | 98.7 | 17 M |
| **APT (fused)** | **0.011±0.01** | **99.0** | 17 M |

[a]MINO exceeds 80 GB GPU memory at the ATES mesh size; see Appendix C.2.1.

*Table 4.* Basin-scale $CO_2$ storage with LGR. APT achieves competitive pressure accuracy using a single unified model with 97.0% fewer parameters than the rigid Nested FNO cascade.

| Model | $\delta\Delta P_g$ | $\delta S_g$ | Params | # of Models |
|---|---|---|---|---|
| FNO-DeepONet | 0.56% | **1.46%** | 96.3M | 5 |
| Nested-FNO | **0.47%** | 1.79% | 682.4M | 5 |
| *APT variants* | | | | |
| APT (GNO only) | 0.93% | 3.9% | **17.7M** | 1 |
| **APT (fused)** | 0.56% | 2.5% | 17.9M | **1** |

As shown in Table 3, APT achieves a relative $L_2$ error of 0.011 (99.0% $R^2$), outperforming interpolated baselines by a factor of 4. This performance gap highlights the penalty of mesh-to-grid conversion, which degrades the sharp thermal fronts that DMO was designed to capture. Among the mesh-native transformer baselines, APT reduces relative $L_2$ error by 2.2× and by 3.6× compared to Transolver and UPT. MINO cannot be evaluated at the ATES mesh size because its latent tensor scales with the input node count, which leads to GPU memory exceeds 80 GB. In terms of computational efficiency over the numerical simulator, APT reduces the computational cost from ∼7 hours for numerical simulation to ∼5 seconds for a full 240-step forecast.

### 3.4. Basin-Scale $CO_2$ Storage with LGR

A essential challenge for $CO_2$ storage simulation is the multi-scale resolution required to capture basin-scale pressure propagation (kilometers) and near-well plume dynamics (meters). Numerical simulators address this via Local Grid Refinement (LGR), which concentrates computational resources near injection wells. However, the resulting mesh hierarchies— involving volume contrasts of 60,000× between coarse and fine cells—create severe difficulties for neural surrogates. The benchmark dataset from Wen et al. (2023b) demonstrates this challenge, featuring five nested refinement levels totaling 0.3–1.0 million cells across a $160 \times 160$ km² domain. More details on the simulation setup are provided in Appendix C.1.3.

Prior state-of-the-art methods handle LGR through hierarchical *model cascades*. For example, Nested FNO (Wen et al., 2023b) trains a separate neural operator for each refinement level. This locks the surrogate to a specific mesh hierarchy; changing the LGR configuration requires retraining the entire cascade. However, APT treats the multi-resolution LGR structure as a single heterogeneous point cloud, learning a unified solution operator end-to-end in a single forward pass (see Figure 9). The details on baseline setup, training details and APT implementations are given in Appendix C.1.3.

**Experiment Results** As detailed in Table 4, APT achieves a global pressure error of 0.56%, comparable to the specialized FNO-DeepONet baseline, while utilizing only 17.9M parameters—a 97.0% reduction compared to Nested FNO's 682.4M. The ablation study on APT variants highlights that the fused gated architecture is particularly effective here. It reduces saturation error from 3.9% to 2.5% and pressure error from 0.93% to 0.56% compared to the non-fused GNO only variant.

Although Nested FNO achieves lower saturation error due to its specialized sub-models, APT offers a superior trade-off between accuracy and flexibility. Furthermore, APT delivers a $6,800\times$ speedup over the ECLIPSE simulator (Schlumberger, 2014), completing full-field inference in 1.4 to 8 seconds compared to 2.75 to 15.93 hours for numerical simulation (see Appendix C.1.3 for more details).

## 4. Discussion

### 4.1. Super-Resolution Evaluation

While mesh convergence is an inherent characteristic of neural operators, recent studies reveal a discrepancy between this promise and practical performance (Sakarvadia et al., 2026; Elrefaie et al., 2025). Here we evaluate APT's super-resolution capabilities on two distinct datasets: CarBench (Elrefaie et al., 2025), which represents smooth aerodynamic pressure fields, and ATES (see Section 3.3). In both cases, models are trained on subsampled data with a fixed node count and subsequently tested on the full-resolution mesh. Detailed experimental set ups, baseline model descriptions, and APT implementation specifics for both CarBench and ATES are provided in Appendix C.1.4 and C.2.1, respectively.

Table 5 illustrates a contrast in behavior that reveals an important distinction between architectural mesh-invariance and training data fidelity. On CarBench, baseline models consistently degraded when evaluated at full resolution, suggesting they overfit to the coarse training grid. In contrast, APT exhibits *mesh-convergent behavior*: performance actually improves by 0.5% as the query resolution increases, confirming that the architecture has learned a continuous operator rather than a discrete mapping tied to the training resolution.

On the subsurface ATES dataset, however, APT shows a modest degradation (1.8%) when evaluated at full resolution. Importantly, this does *not* reflect a limitation of the APT architecture. Rather, this outcome highlights a common challenge in subsurface modeling, which requires extremely high domain size to resolution ratio (6,000× for the ATES dataset (Regnier et al., 2022) and 60,000× for the Nested FNO benchmark dataset). Consequently, fully mesh-converged training data are rare in practice for subsurface

*Table 5.* Super-resolution evaluation. **Top:** On CarBench, APT is the only model to improve at full resolution while baselines degrade; results are reported for a representative unseen test sample (`E_S_WW_WM_648`). **Bottom:** On the heterogeneous ATES dataset, results are averaged over the entire unseen test set.

| CarBench (Smooth) | | |
|---|---|---|
| Model | 10k nodes | Full (~500k) |
| *Baselines* | | |
| NeuralOperator | 83.7 | 81.3 (↓2.4) |
| RegDGCNN | 93.5 | 86.6 (↓6.9) |
| PointTransformer | 94.6 | 86.1 (↓8.5) |
| TripNet | 96.1 | 86.3 (↓9.8) |
| Transolver | 96.5 | 86.6 (↓9.9) |
| AB-UPT | **97.1** | 86.9 (↓10.2) |
| *Ours (Ablation)* | | |
| APT (w/o fused) | 95.1 | 95.2 (↑0.1) |
| **APT (Fused)** | 96.0 | **96.5** (↑0.5) |

| ATES (Heterogeneous) | | |
|---|---|---|
| Model | 4k nodes | Full |
| *Ours (Ablation)* | | |
| APT (w/o fused) | 98.7 | 96.5 (↓2.2) |
| **APT (Fused)** | **99.0** | **97.2** (↓1.8) |

systems, and even an architecturally mesh-invariant model cannot reliably super-resolve beyond the fidelity captured in its training simulations. Super-resolution claims in subsurface applications should be approached with caution, as performance is ultimately bounded by the mesh convergence of the underlying simulation data.

### 4.2. OOD Generalization: Unseen Geostatistical Models

So far, APT is mainly evaluated under in-distribution geo-model settings, where training and testing data share identical underlying statistical properties. In real-world applications, geological priors are often uncertain or incomplete. To investigate strictly *out-of-distribution* (OOD) generalization, we test the model on unseen permeability classes. As shown in Figure 14, the training set contains a mixture of continuous/discontinuous Gaussian and continuous Von Karman fields, while the test set is designed to exclusively include discontinuous von Karman fields—a geomodel regime never seen during training. Detailed experimental setup and baseline configurations are provided in Appendix C.2.2.

Table 15 demonstrates that the APT architecture outperforms baseline approaches in this OOD regime. While standard FNO and ablated APT variants struggle to extrapolate to the unseen discontinuous geostatistical models, APT achieves the lowest relative error (3.28%) and highest $R^2$ (0.91). This demonstrates APT's robustness to the distribution shifts commonly encountered in subsurface applications.

### 4.3. Computational Efficiency

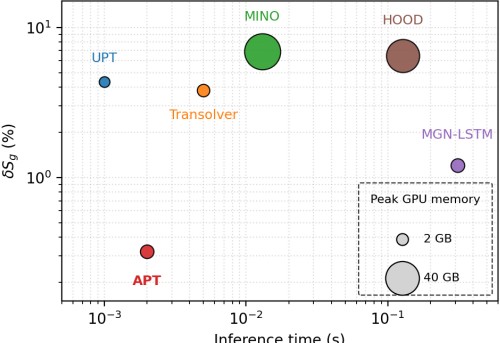

*Figure 4.* Accuracy–efficiency comparison on the 2D faulted $CO_2$ storage benchmark, where each case contains 1024 mesh nodes on average. Inference time is reported for a 19-step rollout using batch size 340 on an NVIDIA H100 80 GB GPU. HOOD is benchmarked on an A100 80 GB GPU with the same batch size and comparable memory capacity. Details of the baseline architectures are summarized in Appendix C.1.2.

We further evaluate the computational efficiency of APT through the accuracy–efficiency comparison shown in Figure 4. Evidently, APT locates at the leading point on the Pareto frontier in the accuracy–inference-time trade-off. APT achieves the lowest saturation error, 0.32%, while maintaining a per-sample inference time of only 2 ms and a peak memory usage of 3.67,GB at batch size 340.

When compared to Transolver, the most competitive transformer-based baseline, APT reduces the saturation error by a factor of nearly 12 (0.32% vs. 3.82%) and accelerates inference by 2.5×, despite operating with a comparable model size. APT also shows clear advantages over more memory-intensive architectures. For example, MINO consumes 46 GB of GPU memory, but yields a saturation error 21.5× higher than APT.

### 4.4. Cross-Dataset Training

A critical requirement for building foundation models in subsurface systems is the ability to intake heterogeneous training data across varying geological structures and mesh configurations. Traditional grid-based approaches (e.g., Nested FNO) are inherently limited by fixed mesh topologies. Such set up prevents training across datasets with different meshes and lead to massive underutilized of public data repositories. For example, the open source basin-scale $CO_2$ storage (Wen et al., 2023b) dataset costs 24,000 20-core CPU hours. However, the dataset can *not* be reused for pre-training if any changed resolution is introduced for a new task. APT's mesh-agnostic architecture inherently enables direct learning from diverse, unstructured point clouds. Here, we investigate whether this capability facilitates pre-training across physically and geometrically distinct domains.

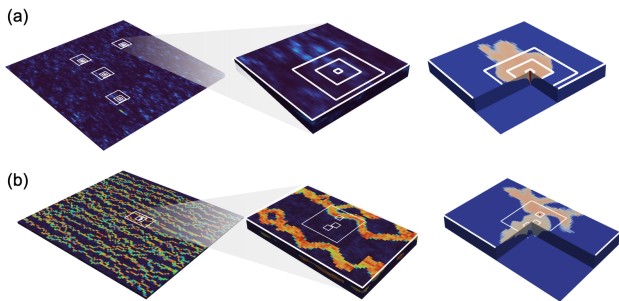

*Figure 5.* Cross-dataset training setup. (a) Gaussian permeability fields defined on nested grids with 4 levels of LGRs. (b) Channelized permeability fields LGR defined with 3 levels of LGRs.

**Experiment Set Up.** We combine two $CO_2$ storage datasets with different semi-adaptive resolutions: (1) a subset of the open source dataset (Wen et al., 2023b) with 4 levels of nested grids, and (2) our created Channelized dataset featuring complex channelized permeability and 3 levels of LGR (Figure 5). This discrepancy in LGR levels makes joint training *impossible* for previous baselines. As a result, one has to train only with the limited Channelized dataset without any additional help from previous published datasets. To demonstrate this, we compare APT's performance when trained on the Channelized set alone versus a joint multi-dataset configuration.

*Table 6.* Cross-dataset training results. Joint training with the Gaussian dataset vs. training only on the Channelized task.

| Training Data | Chan. $R^2$ (%) | Gauss. $R^2$ (%) |
|---|---|---|
| Chan. only (400) | 32.0 | – |
| Gauss. + Chan. (1,400) | **86.7** | **98.9** |

**Results and Implications.** As shown in Table 6, the ability of joint training dramatically improves the performance on the Channalized task with limited data, boosting $R^2$ from 32.0% to 86.7% with a 2.7× improvement.

This experiment demonstrates APT's ability to leverage heterogeneous datasets with incompatible mesh structures. APT enables practitioners to combine previously siloed open source simulation datasets, substantially improving performance on data-limited tasks. This finding suggests a viable path toward pre-training general-purpose subsurface foundation models.

### 4.5. Scaling Behavior

This section examines whether APT exhibits favorable scaling behavior on ATES datasets with respect to the data scale and the model scale, respectively. For data scaling, we adopt the same architecture as used in Section 3.3, and vary the number of training samples from 40k to 120k and 161k.

For model scaling, we use the full 161k-sample training set and increase the model capacity from 0.8M to 17.5M and 69.4M parameters by widening the hidden dimension from 48 to 192 and 384, while keeping the depth fixed. All experiments use the same 10% held-out test split, batch size, optimizer, and learning-rate schedule. We adopt the same training protocol as in Section 3.3 and report the best test loss achieved during training.

*Table 7.* Scaling behavior of APT on ATES. (a) Data scaling at fixed 17.5M parameters; (b) Model scaling at fixed 100% training data (161k samples). Lower is better.

| (a) Data scaling | | (b) Model scaling | |
|---|---|---|---|
| Train samples | Test loss | Parameters | Test loss |
| 40k | 0.1153 | 0.8M | 0.1167 |
| 120k | 0.0829 | 17.5M | 0.0812 |
| 161k | **0.0812** | 69.4M | **0.0722** |

Table 7 shows that APT benefits consistently from both data and model scaling. Increasing the training set from 40k to 161k samples reduces test loss by 29.6%. The improvement continues at 161k, indicating that perfromance gain from increasing data size has not yet saturated on the available data, and more data could futher improve the APT's accuracy. Increasing the parameter count from 0.8M to 69.4M, an 87× scale-up, reduces test loss by 38.1%. Therefore, these results suggest APT's promising role as a scalable backbone for subsurface foundation-model development.

## 5. Conclusion

The APT architecture is a mesh-agnostic neural operator that fuses a Global Perceiver encoder with a Local GNO encoder to resolve the multi-scale nature of subsurface systems. Our results establish APT as a high-performance, scalable backbone for subsurface modeling, characterized by the superior accuracy, direct learning from adaptive mesh simulations, and robust generalization.

One of the main limitations of this work is that we only demonstrate the scaling behavior of APT on single type of subsurface physics. Future study should focus on pre-training on multiple subsurface physics. Also, fine-tuning pretrained APT on new geological domains or physical processes is left as future work.

## Software and Data

This study utilizes a combination of public and synthetic datasets. The APT model architecture and created datasets will be released upon publication.

## Impact Statement

The subsurface is vital for modern society, supplying geothermal energy and minerals while acting as the primary site for carbon sequestration. However, the high cost of traditional simulation hinders effective decision-making. By introducing a model that learns across diverse geological datasets, this work enables a subsurface foundation model for more efficient resource management. This acceleration directly supports the energy transition by providing the speed and accuracy needed to scale carbon storage and renewable geothermal projects.

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

# A. Related Work

## A.1. Convolutional Neural Network (CNN)

CNNs, especially U-Net (Krizhevsky et al., 2012; Ronneberger et al., 2015) were among the first types of architectures adapted for flow prediction tasks in the subsurface. Intuitively, CNNs were used for regular Cartesian discretizations (Zhu & Zabaras, 2018; Mo et al., 2019; Wen et al., 2022; Tang et al., 2022; Wen et al., 2023a; 2021). However, CNN's inherent reliance on fixed computational stencils and structured grid topology restricts their applicability to the unstructured meshes. This is especially problematic for subsurface tasks because it frequently interpolation of complex geological features such as fault discontinuities or stratigraphic pinch-outs (Mallison et al., 2014; Ju et al., 2024). The interpolation of data from unstructured meshes to regular grids for CNN processing can introduce significant discretization errors and potentially miss important physical features resolved by the original computational mesh (Pfaff et al., 2021; Gao et al., 2022; Shukla et al., 2022).

## A.2. Graph Neural Network (GNN)

GNNs provide a natural computational framework for operating directly on unstructured meshes by representing mesh cells/nodes and their connectivity patterns as graph structures (Pfaff et al., 2021; Han et al., 2022; Wu et al., 2022). MeshGraphNet (MGN) have been successfully adopted to model flow behaviors in geometrically complex, faulted reservoir systems (Ju et al., 2024; Jiang, 2024). However, scalability remains a significant limitation for GNNs, as the computational cost and memory requirements of standard message passing algorithms typically scale with the number of nodes and edges. This could easily become prohibitive for industrial-scale simulations involving millions of cells (Li et al., 2020; Alkin et al., 2024; Settgast et al., 2024). Moreover, existing GNN models are often trained autoregressively to account for temporal evolution. Such training method can cause temporal error accumulation over extended prediction horizons and requires mitigation strategies such as controlled noise injection or coupling with recurrent architectures, where the recurrent component can further exacerbate the memory challenge (Lippe et al., 2023; Brandstetter et al., 2022; Sun et al., 2023; Kochkov et al., 2021; Ju et al., 2024).

## A.3. Neural Operators

The neural operator paradigm offers a more theoretically rigorous approach by directly learning the infinite-dimensional-mapping from any functional parametric dependence to the solution (Lu et al., 2021; Li et al., 2021; Bhattacharya et al., 2021; Calvello et al., 2024). Fourier Neural Operators (FNOs) (Li et al., 2021) have demonstrated exceptional performance in GCS modeling applications (Wen et al., 2022; 2023b; Lee et al., 2024). FNOs leverage the computational efficiency of Fast Fourier Transforms (FFTs) to learn global kernel convolutions in the frequency domain, exhibiting robust generalization properties. However, despite the methodological advancement of conducting learning in the function space, standard FNO implementations still assume data defined on regular and pre-defined grids. This leads to complex adaptations for irregular geometries or unstructured meshes. One recent work to mitigate such a limitation is through using a graph-based encoder to map mesh nodes to a latent regular grid (e.g., GINO), where FNO can efficiently operate on (Li et al., 2023).

## A.4. Transformer-based Approaches

Besides the aforementioned CNN, GNNs, and architectures in the neural operators family, transformer-based architectures are expanding rapidly in predicting numerical simulations in recent years (Geneva & Zabaras, 2022; Calvello et al., 2024; Bartolucci et al., 2023; Wang et al., 2024; Li et al., 2022; Lanthaler et al., 2023; Hao et al., 2023). Transformer models are highly effective at finding important connections in sequential data, such as words in a sentence. This ability has led to breakthroughs in many areas, most successfully in language-related tasks (Vaswani et al., 2017; Suk et al., 2024; Wu et al., 2024b; Jaegle et al., 2021). The attention mechanisms can also be interpreted as learning a global integral operator kernel when applied in predicting spatial and temporal domains (Calvello et al., 2024). However, the classical attention operation requires the canonical $\mathcal{O}(N^2)$ computational complexity, where $N$ represents the number of input points/nodes, which prevents direct application to large-scale physical systems.

## A.5. Mesh Adaptivity: R-, H-, and HR-adaptive Refinement

There are three common mesh-adaptivity methods in numerical simulation, which differ in whether the element count and connectivity are changed over time. **R-adaptive** (relocation) moves node positions into high-gradient regions while keeping

the total element count and mesh topology fixed. **H-adaptive** (hierarchical refinement) refines or coarsens the mesh by adding or removing elements, changing the topology. **HR-adaptive** combines both strategies, so node positions, element counts, and connectivity all change between successive time steps.

Most prior mesh-native learning architectures, including MeshGraphNet (Pfaff et al., 2021), Transolver (Wu et al., 2024a), and MINO (Shi et al., 2025), operate on R-adaptive or fixed unstructured meshes; their formulations assume a fixed graph topology shared between input and output. UPT (Alkin et al., 2024) was the first transformer-based architecture able to query input and output on different mesh topologies. Our ATES dataset (Section 3.3) is generated with full HR-adaptive Dynamic Mesh Optimization, producing trajectories whose mesh topology changes at every time step. APT is, to our knowledge, the first architecture demonstrated end-to-end as a learned solution operator on HR-adaptive simulation data.

## B. Architectural Details and Evaluation Metrics

### B.1. Unified Modulation for Temporal and Conditional Inputs

To enable the model to adapt to varying temporal queries and other scalar input conditions and boundary conditions (e.g., $CO_2$ injection rate), we employ a unified modulation strategy.

For any query time $t \in \mathbb{R}$, we first generate a multi-dimensional time embedding $\mathbf{e}_t \in \mathbb{R}^{d_e}$. This can be achieved by multiple encoding methods, and here we use sinusoidal positional encodings, followed by a small multi-layer perceptron (MLP):

$$\mathbf{e}_t = \text{MLP}_{\text{time}}(\text{PE}_{\text{sinusoidal}}(t)) \tag{7}$$

Similarly, other scalar input parameters, such as the injection rate $q_{inj}$, are also embedded into a representation of the same dimension $d_e$ using separate MLPs after appropriate normalization or scaling:

$$\mathbf{e}_{q,t} = \text{MLP}_{\text{injection}}(q_{inj,t}) \tag{8}$$

These individual embedding vectors can then be combined (e.g., by summation or concatenation followed by another MLP) to form a single conditioning vector $\mathbf{e}_{\text{cond},t} \in \mathbb{R}^{d_e}$ that encapsulates both temporal and boundary condition information.

This combined conditioning embedding $\mathbf{e}_{\text{cond},t}$ then modulates all the transformer blocks within the Encoder ($\mathcal{E}$), Approximator ($\mathcal{A}$), and Decoder ($\mathcal{D}$), as discussed above. We adopt a modulation mechanism by following Diffusion Transformers (Peebles & Xie, 2023).

### B.2. Evaluation Metrics

To rigorously evaluate model performance, we employ a set of well-established metrics that quantify prediction accuracy, error, and generalization capabilities.

**Coefficient of Determination** ($R^2$). The $R^2$ metric quantifies the proportion of variance in the ground truth that is explained by the model's predictions:

$$R^2 = 1 - \frac{\sum_{i=1}^{N}(z_i - \hat{z}_i)^2}{\sum_{i=1}^{N}(z_i - \bar{z})^2}, \tag{9}$$

where $z_i$ denotes the ground truth value, $\hat{z}_i$ is the predicted value, $\bar{z}$ is the mean of the ground truth, and $N$ is the number of samples.

**Relative L2 Error.** The relative L2 error evaluates the normalized L2 error between predictions and ground truth:

$$\text{Relative L2} = \frac{\|\hat{z} - z\|_2}{\|z\|_2}. \tag{10}$$

**Saturation Plume Error** ($\delta S$). To quantify prediction accuracy for phase saturation, we use the plume saturation error introduced in Wen et al. (2023a). This metric can be computed for any phase (e.g., gas, oil, or water) using the same

formulation; here we illustrate with the gas phase ($\delta S_g$):

$$\delta S_g = \frac{1}{\sum_{i,n} I_i^n} \sum_{n=1}^{N_T} \sum_{i=1}^{N_C} I_i^n |S_{g,i}^n - \hat{S}_{g,i}^n|,$$

$$I_i^n = 1 \quad \text{if} \quad (S_{g,i}^n > 0.01) \cup (\hat{S}_{g,i}^n > 0.01),$$

(11)

where $I_i^n = 1$ indicates that a mesh cell has non-zero saturation in either the ground truth or the prediction, $S_{g,i}^n$ denotes the ground truth gas saturation, $\hat{S}_{g,i}^n$ is the predicted gas saturation, $N_T$ is the number of temporal snapshots, and $N_C$ is the number of mesh cells. This metric focuses on accuracy within the plume region.

**Relative Pressure Error ($\delta P$).** The relative pressure error quantifies the normalized pressure prediction error. This metric can be computed for any phase; here we illustrate with the gas phase ($\delta P_g$):

$$\delta P_g = \frac{1}{N_C N_T} \sum_{n=1}^{N_T} \sum_{i=1}^{N_C} \frac{|P_{g,i}^n - \hat{P}_{g,i}^n|}{P_{g,\text{init}}},$$

(12)

where $P_{g,i}^n$ is the ground truth pore pressure, $\hat{P}_{g,i}^n$ is the predicted pressure, and $P_{g,\text{init}}$ is the initial reservoir pressure.

**Pressure Buildup Error ($\delta \Delta P$).** For benchmarks predicting pressure buildup ($\Delta P$) directly, the error metric follows the same phase-agnostic formulation. Using gas phase as an example:

$$\delta \Delta P_g = \frac{1}{N_C N_T} \sum_{n=1}^{N_T} \sum_{i=1}^{N_C} \frac{|\Delta P_{g,i}^n - \Delta \hat{P}_{g,i}^n|}{\Delta P_{g,\text{max}}},$$

(13)

where $\Delta P_{g,\text{max}}$ is the maximum ground truth pressure buildup over all cells and time steps. This metric is commonly used for evaluating reservoir pressure buildup (Tang et al., 2022; Wen et al., 2022).

## B.3. Training Cost Summary

The primary drivers of GPU memory consumption for APT are: (1) the number of input points sampled per training step, (2) the number of supernodes used for global attention, and (3) the latent dimension. As summarized in Table 8, per-epoch training time scales with both dataset size and mesh complexity. Table 9 provides additional efficiency metrics including throughput and memory utilization per million parameters.

*Table 8.* APT Training System Performance Summary. Memory/GPU shows per-device memory usage with utilization percentage in parentheses.

| Dataset | Model Size | Input Cells | Batch Size | GPU Setup | Memory/GPU | Total Memory | Time/Epoch | Samples/Epoch |
|---|---|---|---|---|---|---|---|---|
| Faulted | 1.2 M | 1,024 | 32 | 1×40GB[a] | 2.8 GB (7%) | 2.8 GB | ~23 sec | 8,544 |
| Hydrocarbon | 12 M | 32,768 | 32 | 1×80GB[b] | 43.8 GB (55%) | 43.8 GB | ~10.5 min | 13,984 |
| Car Bench | 6.70 M | 10,000 | 8 | 8×40GB[a] | 16.8 GB (42%) | 134.4 GB | ~20 sec | 728 |
| ATES | 17 M | 8,192 | 16 | 8×40GB[a] | 5.6 GB (14%) | 44.8 GB | ~27 min | 20,160 |
| Basin-scale GCS | 17.9 M | 262,144 | 4 | 8×80GB[b] | 23.8 GB (30%) | 190 GB | ~55 min | 7,224 |

[a]NVIDIA A100 40GB SXM    [b]NVIDIA A100 80GB SXM

*Table 9.* APT Training Efficiency Metrics.

| Dataset | Model Size | Input Cells | Throughput (samples/sec) | Effective Batch | Memory/M Params (GB) |
|---|---|---|---|---|---|
| Faulted | 1.38 M | 1,024 | ~371 | 32 | 2.33 |
| Hydrocarbon | 12 M | 32,768 | ~22 | 32 | 3.65 |
| Car Bench | 6.70 M | 10,000 | ~36 | 64 | 2.51 |
| ATES | 17 M | 8,192 | ~12 | 128 | 0.33 |
| Basin-scale GCS | 17.9 M | 262,144 | ~2.2 | 32 | 1.33 |

# C. Benchmark Datasets

We evaluate APT on seven benchmark datasets spanning structured grids, unstructured meshes, and adaptive meshes. The first four are published datasets that enable direct comparison with prior work; the remaining three are newly created to evaluate APT on adaptive meshes, out-of-distribution generalization, and multi-dataset training.

## C.1. Published Datasets

### C.1.1. CARTESIAN GRID HYDROCARBON EXTRACTION

Hydrocarbon extraction is a subsurface engineering process focused on the recovery of oil and gas resources from porous geological formations to meet global energy demands (Badawi & Gildin, 2025; Aziz, 1979). The operation relies on complex multiphase fluid dynamics governed by Darcy's law, where injected fluids are often used to displace hydrocarbons through heterogeneous media. Accurate reservoir management requires extensive numerical simulation for history matching and production optimization to mitigate high geological uncertainty. However, given the large scale of reservoir models and the need for thousands of realizations, conventional high-fidelity simulators often become computationally prohibitive for multi-scenario studies.

**Governing Equations.**     Two-phase oil-water flow in porous media is governed by the mass balance equation combined with Darcy's law:

$$\frac{\partial(\phi\rho_j S_j)}{\partial t} - \nabla \cdot [\rho_j \lambda_j \mathbf{K}(\nabla p_j - \rho_j g \nabla D)] + q_j^{ss} = 0, \tag{14}$$

where subscript $j \in \{w, o\}$ denotes the phase (water or oil), $\rho_j$ is phase density, $\phi$ is porosity, $\mathbf{K}$ is the permeability tensor, and $\lambda_j = k_{rj}/\mu_j$ is phase mobility with relative permeability $k_{rj}$ and viscosity $\mu_j$. The term $q_j^{ss}$ represents sources/sinks (wells). The system is closed with saturation constraint $S_w + S_o = 1$ and capillary pressure $p_c(S_w) = p_o - p_w$.

**Dataset Description.**     The dataset from Badawi & Gildin (2025) simulates two-phase flow on a $40 \times 40$ Cartesian grid using the CMG IMEX simulator (CMG, 2014). Each sample has a unique heterogeneous Gaussian permeability field. The simulation spans 10 years with 10-day timesteps (366 snapshots total). Training samples contain 6 wells (4 producers, 2 injectors) with random locations and controls; test samples vary from 3 to 11 wells. Following (Badawi & Gildin, 2025), we adapt the data split ratio as follows. Training: 3,600 samples (augmented to 14,400 via flipping); Validation: 400 samples; Testing: 200 samples (varying well count: 3–11 wells).

**Learning Problem.**     Given input $a(\mathbf{x}) = [\mathbf{K}(\mathbf{x}), \text{well locations, well controls}, P_0(\mathbf{x}), S_{w,0}(\mathbf{x})]$, the model predicts pressure $P(\mathbf{x}, t)$ and water saturation $S_w(\mathbf{x}, t)$ for 40 consecutive timesteps (400 days) starting from any initial condition. Training uses timesteps $[0, 2390]$ days; testing extends to $[0, 3650]$ days to evaluate temporal extrapolation.

**APT Implementation and Training Details.**     At each training step, input grids are sampled with a fixed size of 32,768 cells, around 50% of the original cell counts. We employ a multi-branch APT architecture to jointly predict pressure and saturation fields. Each branch contains a fused encoder combining local message-passing pooling with global cross-attention pooling, followed by four self-attention blocks and a perceiver module that compresses to 512 latent tokens. The two branch representations are fused via a two-layer transformer before being processed by separate approximators and decoders. The model uses a hidden dimension of 192, 4 attention heads, and an MLP expansion ratio of 4, totaling 12.7M parameters. The model is trained for 600 epochs with a batch size of 32, corresponding to 13,984 samples per epoch. We use the AdamW optimizer with a learning rate of $1 \times 10^{-4}$ and weight decay of $10^{-4}$. A cosine learning rate schedule is employed with a warmup phase of 20 epochs. Training is conducted on a single NVIDIA A100 80GB SXM GPU, utilizing approximately 43.8 GB of memory (55% utilization) with $\sim$10.5 minutes per epoch. The more complete training cost of this dataset is summarized in Table 8.

**Baselines.**     FNO and U-FNO results are from Badawi & Gildin (2025). Both baselines use width 64, 10 Fourier modes, and are trained with relative $L_2$ loss including gradient terms.

### C.1.2. IRREGULAR GRID CO$_2$ STORAGE

**Governing Equations.**     We consider a multi-phase flow problem with $CO_2$ and water in the context of geological storage of $CO_2$. The system's dynamics are described by miscible two-phase (gas and aqueous) two-component ($H_2O$ and $CO_2$)

flow in a compressible porous medium. The general forms of mass accumulations for component $\eta = CO_2$ or $water$ are written as (Pruess, 2005):

$$\frac{\partial}{\partial t}\Big(\varphi \sum_p S_p \rho_p X_p^{CO_2}\Big) = -\nabla \cdot \mathbf{F}_{adv}^{CO_2} + q^{CO_2}, \tag{15}$$

$$\frac{\partial}{\partial t}\Big(\varphi \sum_p S_p \rho_p X_p^{water}\Big) = -\nabla \cdot \mathbf{F}_{adv}^{water}. \tag{16}$$

Here $p \in \{a, g\}$ indexes the aqueous ($g$) and gas ($g$) phases; in this work, the aqueous phase is water while the gas phase is $CO_2$ (Pini et al., 2012). The symbols $\varphi$, $S_p$, $\rho_p$ and $X_p^\eta$ denote, respectively, porosity, phase saturation, mass density, and mass fraction of component $\eta$ in phase $p$; $q^{CO_2}$ represents external sources or sinks of $CO_2$. For either component, the advective flux equals the phase-weighted sum

$$\mathbf{F}_{adv}^\eta = \sum_p X_p^\eta \mathbf{F}_p, \qquad \mathbf{F}_p = -k \frac{k_{r,p}\rho_p}{\mu_p}\big(\nabla P_p - \rho_p \mathbf{g}\big),$$

where $\mathbf{F}_p$ follows the multiphase extension of Darcy's law. Absolute permeability $k$ is a rock property, while relative permeability $k_{r,p}(S_p)$ and viscosity $\mu_p(P_p)$ are highly non-linear phase functions. Gravitational effects are captured by $\mathbf{g}$. Capillarity introduces a pressure offset between phases,

$$P_g = P_a + P_c(S_p), \qquad P_a = P_a,$$

with $P_c(S_p)$ a saturation-dependent capillary pressure function. Consequently $\varphi$, $\rho_p$, and the solubility terms $X_p^\eta$ in Eqs. (15)–(16) are implicit functions of the phase pressures $P_p$.

**Finite-Volume Discretization.** The governing equations (15)–(16) are discretized with a cell-centered, fully implicit (backward-Euler) finite-volume scheme based on a two-point flux approximation (TPFA) and single-point upstream weighting. The primary variables are chosen to be the gas-phase pressure $P_g$ and the overall component densities $\rho_{H_2O}$ and $\rho_{CO_2}$, where an overall component density represents the mass of a given component per unit volume of mixture (Voskov & Tchelepi, 2012). At each time step, the nonlinear system of discretized equations is solved with Newton's method with damping to update all primary variables in a fully coupled fashion.

**PEBI Mesh Generation.** Unstructured polygonal meshes are well suited to represent complex faults and to perform local spatial refinement around injection wells. We use perpendicular bisector (PEBI) grids (Palagi & Aziz, 1994; Lie, 2016) generated with the MATLAB Reservoir Simulation Toolbox (MRST) to mesh a $1\,km \times 1\,km \times 1\,m$ domain containing an injector well and two straight impermeable faults that are conformal with the grid. Since PEBI meshes are orthogonal by construction, the flow simulations can be performed with a TPFA-based finite-volume scheme without compromising solution accuracy.

All simulations are performed with GEOS (Settgast et al., 2024), an open-source multiphysics simulator for geological carbon storage. The domain is initially saturated with brine at $10\,MPa$ and $143.76°C$. Supercritical $CO_2$ is injected at $0.058\,kg/s$ for 950 days, with analytical (Carter-Tracy) aquifer boundary conditions.

**Dataset Details.** All input and output fields are normalized using z-score scaling, and simulation timesteps are scaled to $[0, 1]$. The z-score normalization is defined as:

$$\tilde{\mathbf{x}}_i^n = \frac{\mathbf{x}_i^n - \text{mean}\big([\mathbf{x}_1^n, \ldots, \mathbf{x}_{n_S}^n]\big)}{\text{std}\big([\mathbf{x}_1^n, \ldots, \mathbf{x}_{n_S}^n]\big)}, \quad i = 1, \ldots, n_S, \quad n = 1, \ldots, n_T \tag{17}$$

The input and output features for all machine learning models are summarized in Table 10. Figure 6 illustrates three randomly selected geomodel realizations, showing the heterogeneous permeability fields with two fixed impermeable faults and varying injection well locations.

*Figure 6.* Heterogeneous permeability realizations with two fixed impermeable faults and one injection well for three cases. The well coordinates for each case are shown at the top, with insets displaying an enlarged view of the well vicinity.

*Table 10.* Input and output of the ML model: $s_{g,i}^n, V_i, k_i, \boldsymbol{n}_i, \boldsymbol{x}_i$ denote gas saturation, cell volume, scalar permeability, cell type, cell center, respectively, for a given cell. Here, $\boldsymbol{n}_i$ is a one-hot vector of size 4, encoding whether a cell is an internal cell, injector, cell along fault lines, and boundary cell.

| Case | Node input | Edge input | Node output |
|------|-----------|-----------|-------------|
| Baseline | $s_{g,i}^n, V_i, k_i, \mathbf{n}_i, \mathbf{x}_i$ | $\mathbf{x}_i - \mathbf{x}_j,$ $\|\mathbf{x}_i - \mathbf{x}_j\|$ | $s_{g,i}^{n+1}$ |

**APT Implementation and Training Details** During each training step, input grids are sampled with a fixed size of 1024 cells. The APT model has a total of 1.38M parameters. The architecture consists of an encoder, an approximator, and a decoder, each containing four perceiver-style attention layers, resulting in twelve blocks for the core transformer. The model uses a hidden dimension of 48, 128 latent tokens, and 3 attention heads per block.

A hybrid positional encoding strategy is employed: sinusoidal encoding is used for time and scalar inputs, whereas learnable embedding tables with interpolation are used for spatial coordinates. All spatial coordinates are rescaled to the range $[0, 200]$ to facilitate the positional encoding computation. For fused encoder, the model constructs 256 supernodes for each case using both a graph pooling with a radius of 20.0 (a quarter of the scaled simulation domain) and a perceiver layer for global pooling. The perceiver module in the encoder compresses the encoded features to 128 latent tokens. The model uses a hidden dimension of 48, and a conditioning dimension of 192.

The model is trained for 200 epochs with a batch size of 64. We use the Lion optimizer (Chen et al., 2023) with a learning rate of $1 \times 10^{-4}$ and a weight decay of 0.5. A cyclic learning rate schedule is employed, with a warmup fraction of 0.2, where the learning rate is annealed over the training duration. We train all DL models using an Nvidia A100-SXM GPU.

**Baselines.** We compare APT against graph-based and grid-based baselines. For graph-based baselines, we adopt Mesh-GraphNet (MGN) and MGN-LSTM (Ju et al., 2024) (previously SOTA on this dataest) to operate directly on the unstructured mesh without interpolation. MGN-LSTM is trained autoregressively on 19-step sequences, while MGN is trained for single-step prediction; both generate the full 950-day sequence via autoregressive rollout at inference. We use the checkpoints and noise injection strategy from Ju et al. (2024).

U-Net (Ronneberger et al., 2015) and U-FNO (Wen et al., 2022), however, require structured grid inputs. Since the PEBI mesh is unstructured with varying topology across cases, the simulation data must be interpolated onto a regular grid before training. As the average cell number of this dataset is around 1600, we use a resolution of $40 \times 40$ to interpolate the field using linear interpolation with nearest-neighbor filling at boundaries. However, this interpolation introduces approximation error at cell boundaries and near faults where the mesh is locally refined. Both models are trained as direct predictors in a fashion similar to how APT is trained to ensure fairness. Namely, the model takes the initial state concatenated with a normalized time encoding to directly predict the output state at the target time, rather than using autoregressive rollout.

The U-Net uses 4 encoder and 4 decoder blocks with skip connections and a base channel width of 18. U-FNO employs four Fourier layers with 12 modes in each spatial dimension, a hidden dimension of 32, and incorporates U-Net blocks in the third and fourth layers for multi-scale feature extraction. Both models have comparable parameter counts ($\sim$1.4M) for fair comparison: U-FNO with 1.48M parameters and U-Net with 1.36M parameters. All grid-based models are trained with batch size 16 for 100 epochs using AdamW optimizer with cosine learning rate scheduling.

**Transformer Baselines (UPT, Transolver, MINO).** We additionally compare against three transformer-based operator baselines, all trained on the same 2D $CO_2$ storage dataset with direct one-step prediction. UPT (Alkin et al., 2024) uses

a local supernode encoder ($\dim=192$, 4 layers, 1024 supernodes, 128 latent tokens) followed by global self-attention; 1.68M parameters. Transolver (Wu et al., 2024a) applies physics-aware token slicing followed by global self-attention ($n_{\text{hidden}}=128$, 8 layers, 8 heads, 32 slices); 1.59M parameters. MINO (Shi et al., 2025) uses a graph-neural-operator encoder with a continuous kernel integral transform at each point pair within a fixed radius ($n_{\text{hidden}}=128$, 4 GNO layers); 1.64M parameters. All three baselines are trained for 200 epochs with AdamW (learning rate $10^{-3}$, weight decay $10^{-5}$) on identical train/test splits, with the same evaluation metrics as APT.

**Hierarchical Graph Baseline (HOOD).** A hierarchical graph baseline, HOOD (Grigorev et al., 2023), is also considered to compare against APT. HOOD uses an encoder–processor–decoder pipeline with farthest-point-sampling coarsening across three hierarchy levels and U-Net-style graph message passing; tokens at every level are purely local and gain wider receptive fields only through progressive coarsening, rather than through joint local–global encoding as in APT. We adapt the public architecture to the Faulted 2D $CO_2$ dataset with matched parameter count ($\sim$2.0M; latent dimension 112, 3 levels, 3 message-passing steps per level) and train for 400 epochs with AdamW (learning rate $10^{-3}$, weight decay $10^{-5}$).

**Additional Results.** Here, we consider 5 representative meshes from the test set to demonstrate that APT generalizes well for different meshes, boundary conditions (well locations), and permeability fields that are not included in the training set. The predicted results after 950 days are presented in Figure 7 for saturation and Figure 8 for pressure, respectively. Due to the complex interplay between the injection front and the faults, the differences in initial setup yield very different outcomes.

Across all five cases, APT's predictions (third row) for both saturation and pressure match greatly with the ground truth simulations (second row). The corresponding prediction errors (fourth row) are consistently low for both fields. This observation confirms that APT generalizes effectively to new geological realizations and well locations, greatly outperforming the MGN-LSTM model (bottom row) in both prediction tasks.

Table 11 presents the inference time required for a 19-step prediction (equivalent to 950 simulation days) on the same Faulted 2D $CO_2$ benchmark, measured at batch size 340 on an NVIDIA H100 80 GB GPU. The high-fidelity simulator, GEOS, takes 49.02 seconds on a single-core CPU. Among the neural baselines, the graph-based predictors MGN-LSTM and standard MGN complete the prediction in 0.31 s and 0.07 s; the transformer-based baselines Transolver and UPT reach 0.005 s and 0.001 s; MINO and the hierarchical-graph HOOD baseline run in 0.013 s and 0.128 s, respectively. APT completes the same prediction in 0.002 s, while attaining the lowest saturation error in the comparison (Figure 4). UPT is slightly faster ($44{,}939\times$), but has an order-of-magnitude higher saturation error.

*Table 11.* Inference time and relative speedup for a 19-step rollout (950 days) on the Faulted 2D $CO_2$ benchmark, measured at batch size 340.

| Model | Time (s) | Speedup |
|---|---:|---:|
| GEOS[a] | 49.02 | $1\times$ |
| MGN-LSTM[b] | 0.31 | $158\times$ |
| HOOD[c] | 0.128 | $383\times$ |
| Standard MGN[b] | 0.07 | $700\times$ |
| MINO[b] | 0.013 | $3{,}668\times$ |
| Transolver[b] | 0.005 | $9{,}593\times$ |
| **APT**[b] | **0.002** | **19,873$\times$** |
| UPT[b] | 0.001 | $44{,}939\times$ |

[a] Intel Xeon E5-2695 v4, single-core serial run. [b] NVIDIA H100 80 GB GPU. [c] NVIDIA A100 80 GB GPU.

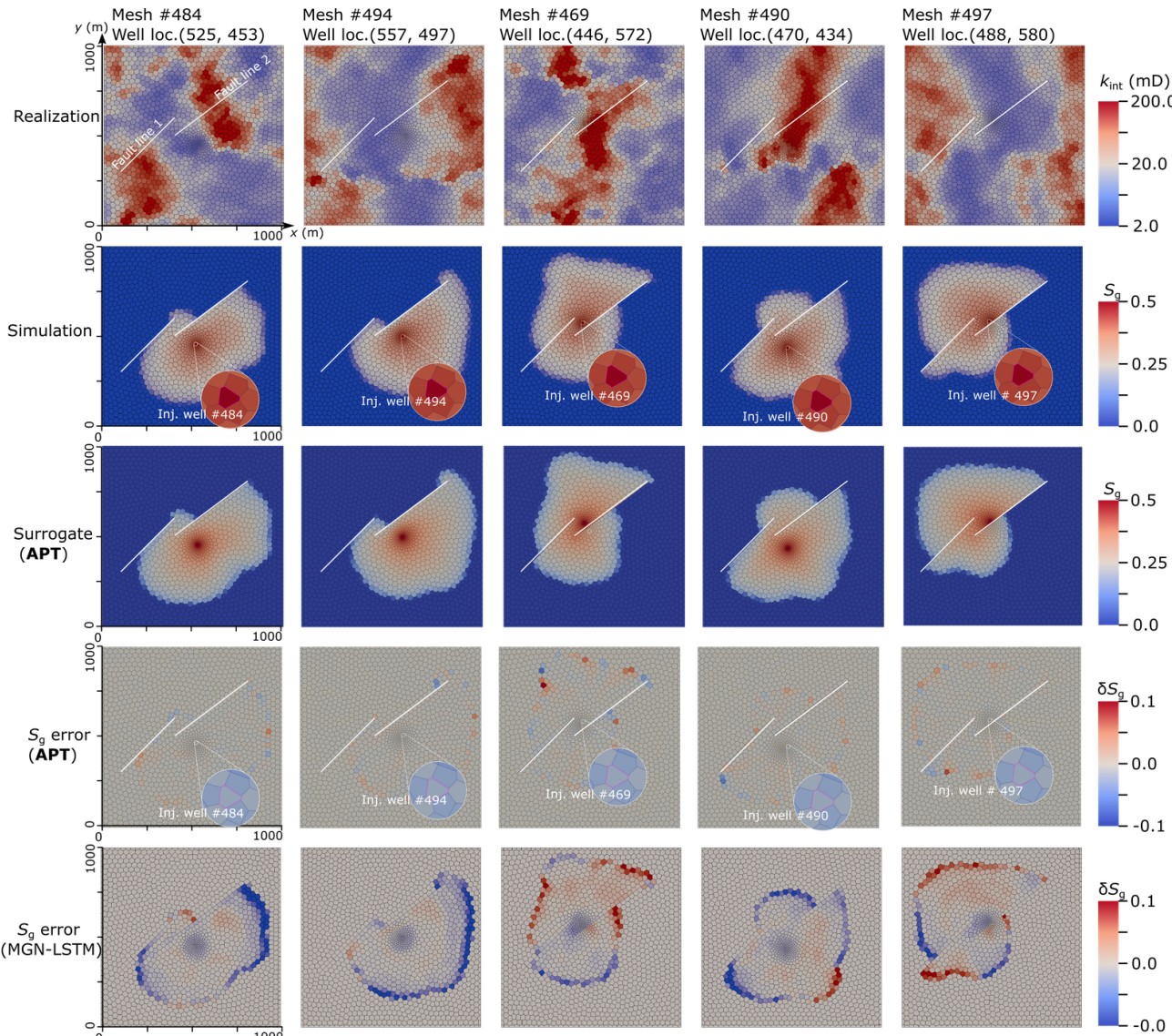

*Figure 7.* Comparison of model generalizability for gas saturation predictions. Predictions at 950 days are shown across five distinct test meshes with varied permeability fields and well locations. The rows, from top to bottom, display: (1) reservoir permeability, (2) the high-fidelity (HF) simulation ground truth, (3) the APT prediction, (4) the APT prediction error, and (5) the MGN-LSTM prediction error.

### C.1.3. 3D Nested Local Grid Refined (LGR) Dataset

**Dataset Details.** The dataset comprises 3,011 ECLIPSE (e300) simulations (Schlumberger, 2014) of $CO_2$ injection over 30 years into dipped 3D reservoirs. Each simulation covers a domain of 160 km $\times$ 160 km $\times$ 100 m and includes one to four injection wells. A key characteristic is its hierarchical multiresolution meshing, where Local Grid Refinements (LGRs) are applied around injection wells to capture high-gradient dynamics, while coarser grids model far-field regions.

The grid structure features five refinement levels, from a coarse global mesh (Level 0, 50,000 cells) to the finest mesh near wells (Level 4, 80,000 cells). This results in total mesh sizes ranging from approximately 0.3 to 1.0 million cells per simulation. The dataset also covers a broad range of physical and operational parameters, including reservoir depth (800–4500m), dip angle (0–2°), and diverse injection schemes and permeability structures. Full sampling distributions are provided in Table 12.

Each of the 3,011 simulations includes 24 irregularly spaced time snapshots, yielding 72,264 total space-time instances. Following Wen et al. (Wen et al., 2023a), the data is randomly split into training, validation, and test sets using an 8:1:1 ratio.

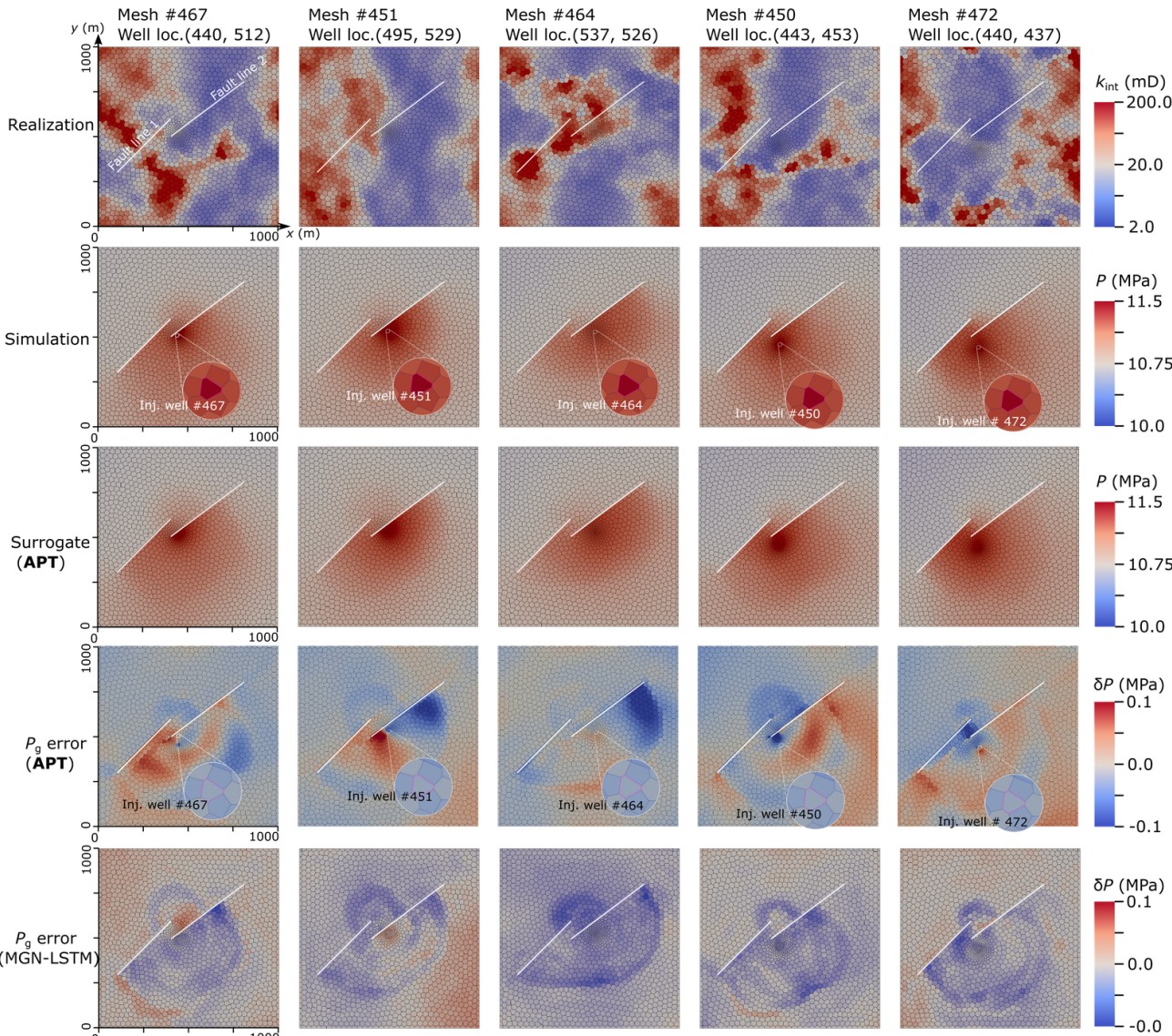

*Figure 8.* Comparison of model generalizability for pore pressure predictions. Predictions at 950 days are shown across five distinct test meshes with varied permeability fields and well locations. The rows, from top to bottom, display: (1) reservoir permeability, (2) the high-fidelity (HF) simulation ground truth, (3) the APT prediction, (4) the APT prediction error, and (5) the MGN-LSTM prediction error.

Training proceeds for up to 300 epochs with early stopping based on validation error.

The sampling parameters and distributions for the reservoir simulation variables are detailed in Table 12, following the dataset description in (Wen et al., 2023a). For a comprehensive overview of the simulation setup, please refer to (Wen et al., 2023b).

**APT Implementation.** At each training step, input grids are sampled with a fixed size of 262,144 cells. Although the number of cells varies across simulation cases in the Nested LGR dataset, using a fixed sample size ensures stable memory usage. The APT model has a total of 17.9M parameters. Its architecture includes an encoder, an approximator, and a decoder, each with four perceiver-style attention layers, totaling twelve blocks. The model uses a hidden dimension of 192, 8192 supernodes, 1024 latent tokens, and 4 attention heads per block to balance accuracy and efficiency.

Similar to the 2D experiments in Section C.1.2, we use sinusoidal embedding for time and scalar inputs and learnable embeddings for spatial coordinates. For information aggregation, the model constructs 4096 supernodes for each case using

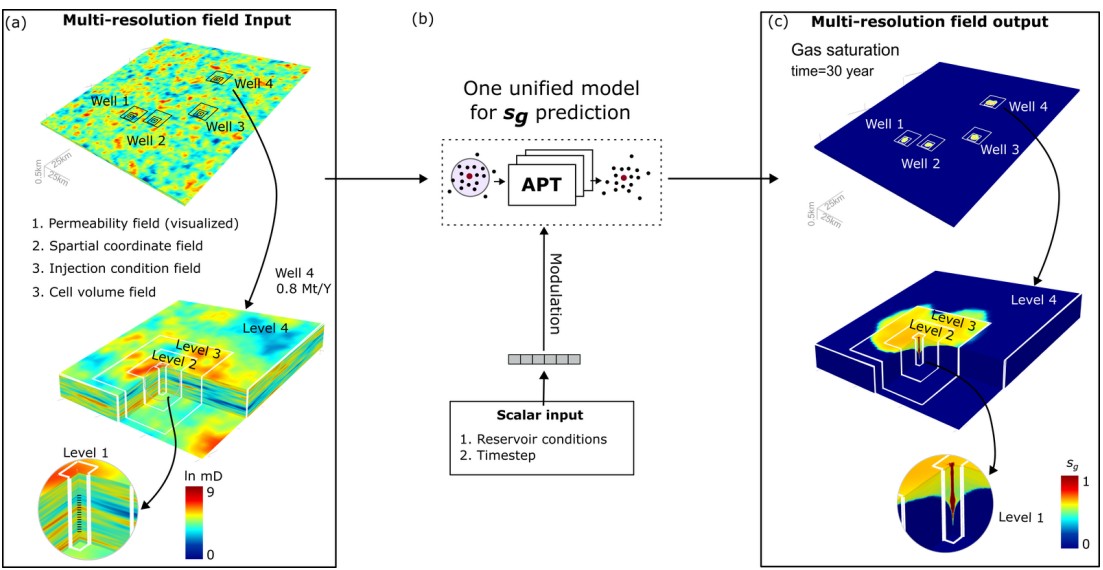

*Figure 9.* APT framework for multiresolution CO$_2$ storage modeling. (a) Input fields include heterogeneous permeability, spatial coordinates, injection conditions (e.g., Well 4), and cell volumes across LGR levels. (b) A single, unified APT model processes all multiscale fields and scalar parameters end-to-end. (c) The model outputs predictions, such as the gas saturation field at year 30, visualized across all refinement levels. Adapted from Wen et al. (Wen et al., 2023b).

*Table 12.* Sampling parameters and distributions for reservoir simulation variables (adapted from Wen et al. (Wen et al., 2023b))

| Variable type | Sampling parameter | Distribution | Unit |
|---|---|---|---|
| Permeability map | $x$-axis correlation | $X \sim \mathcal{U}[800, 4000]$ | m |
| | $y$-axis correlation | $X \sim \mathcal{U}[800, 4000]$ | m |
| | $z$-axis correlation | $X \sim \mathcal{U}[4, 20]$ | m |
| | Reservoir permeability mean | $X \sim \mathcal{U}[4.09, 5.01]$ | ln mD |
| | Reservoir permeability std | $X \sim \mathcal{U}[0.25, 1]$ | ln mD |
| Reservoir conditions | Reservoir center depth | $X \sim \mathcal{U}[800, 4500]$ | m |
| | Geothermal gradient | $X \sim \mathcal{U}[15, 35]$ | °C/km |
| | Dip angle | $X \sim \mathcal{U}[0, 2]$ | ° |
| Injection design | Number of wells | $n \in \{1, 2, 3, 4\}$ | – |
| | Injection rate | $X \sim \mathcal{U}[0.5, 2]$ | MT/y |
| | Perforation thickness | $X \sim \mathcal{U}[20, 100]$ | m |
| | Perforation location | Randomly placed | – |

a fixed-radius sampling strategy with a radius of 20.0 (one-tenth of the scaled simulation domain). The maximum number of neighboring cells in each graph pooling operation is capped at 128.

**Hyperparameters.** We train the model for 300 epochs with a batch size of 8, including a warmup phase of 60 epochs. Hyperparameters were selected via a small grid search on the validation set. Importantly, we found that a relatively small number of supernodes (8192 for this dataset) suffices to achieve good performance. We train both APT and baseline models using an Nvidia A100-SXM GPU. For APT, each epoch takes roughly 55 minutes. For optimization, we use the Lion optimizer (Chen et al., 2023) with a learning rate of $1 \times 10^{-4}$ and a weight decay of 0.05.

**Baselines.** The Nested FNO model from (Wen et al., 2023a) is used as a baseline. For the comparisons, we utilize the checkpoints provided by the authors. For further details, please refer to the original paper (Wen et al., 2023a).

**Computational Efficiency.** To further assess runtime scalability, Table 13 compares the end-to-end runtime for APT and ECLIPSE simulations under different injection well configurations. ECLIPSE simulations are executed using 20-core Intel Xeon E5-2640 CPUs, while APT and Nested FNO are evaluated on Nvidia A100-SXM GPUs. APT consistently achieves more than $6,800\times$ speed-up relative to ECLIPSE across all tested scenarios.

*Table 13.* Runtime comparison between ECLIPSE simulator (HF) and APT across different well configurations. Speed-up is computed as the ratio of ECLIPSE runtime to APT runtime.

| # Wells | # Cells | ECLIPSE[a] (hr) | APT[b] (s) | APT Speedup |
|---|---|---|---|---|
| 1 | 296,300 | 2.75 | 1.44 | 6,875 |
| 2 | 542,600 | 6.43 | 3.40 | 6,808 |
| 3 | 788,900 | 11.00 | 5.48 | 7,226 |
| 4 | 1,035,200 | 15.93 | 8.00 | 7,168 |

[a] Intel Xeon E5-2640 CPUs.      [b] NVIDIA A100 GPU.

**Additional Results.** As shown in Figure 10, the model accurately captures the overall plume evolution patterns across the three wells with different injection rates (1.19, 1.87, and 2.20 MT/y) after injecting for 30 years. The predictions show good agreement with the reference solutions, with some localized errors observed at plume boundaries and high-gradient regions where saturation transitions occur rapidly.

The temporal predictions for Well 2 shown in Figure 11 further reveal important insights into the model's performance across different time scales and spatial resolutions. During the earlier stages (1-10 years), the plume saturation error remains well constrained, with the $CO_2$ plume primarily contained within the first two local grid refinements (LGRs). However, plume prediction errors begin to exacerbate as the plume migrates into the outer LGR 3 region by 30 years, where the mesh resolution is significantly coarser.

### C.1.4. CarBench: 3D Car Aerodynamics Benchmark

**Dataset description.** CarBench (Elrefaie et al., 2025) is a large-scale benchmark for neural surrogate modeling of 3D car aerodynamics, built on the DrivAerNet++ dataset (Elrefaie et al., 2024b). The dataset comprises 8,150 parametrically generated car configurations across three body types (Fastback, Estateback, Notchback), each simulated using steady-state Reynolds-Averaged Navier-Stokes (RANS) equations with OpenFOAM. Surface meshes contain approximately 500,000 points per car, with the task being to predict surface pressure fields from geometry alone.

**Training and evaluation protocol.** Following the CarBench protocol, all models are trained on 10,000 uniformly sampled surface points per geometry. Evaluation is conducted at both the subsampled resolution (10k points) and full mesh resolution ($\sim$500k points). This dual evaluation reveals whether models learn resolution-invariant representations or suffer from aliasing artifacts when queried at higher resolutions.

**Implementation details.** We utilize the same APT architecture described in Appendix C.1.3, omitting the temporal modulation component to align with the steady-state nature of the Carbench dataset. Specifically, we set the latent dimension to $d_h = 192$. The network depth is configured with 2 transformer blocks in the encoder, 2 blocks in the approximator, and 4

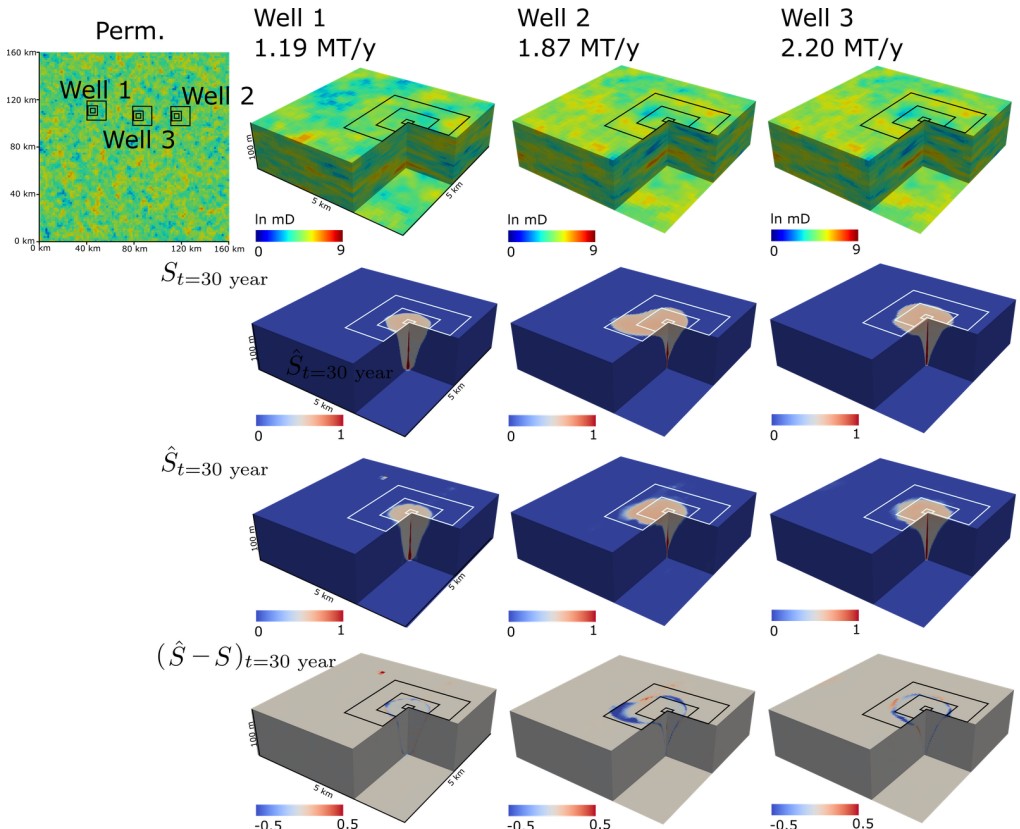

*Figure 10.* APT model's predictions for gas saturation fields at year 30 from a 3-well setting. Each row corresponds to a different LGR resolution level, and each column shows a different test case. From top to bottom: (1) Permeability fields and LGR meshes, (2) APT-predicted gas saturation ($S_g$), (3) Ground truth saturation from ECLIPSE simulations, and (4) prediction errors.

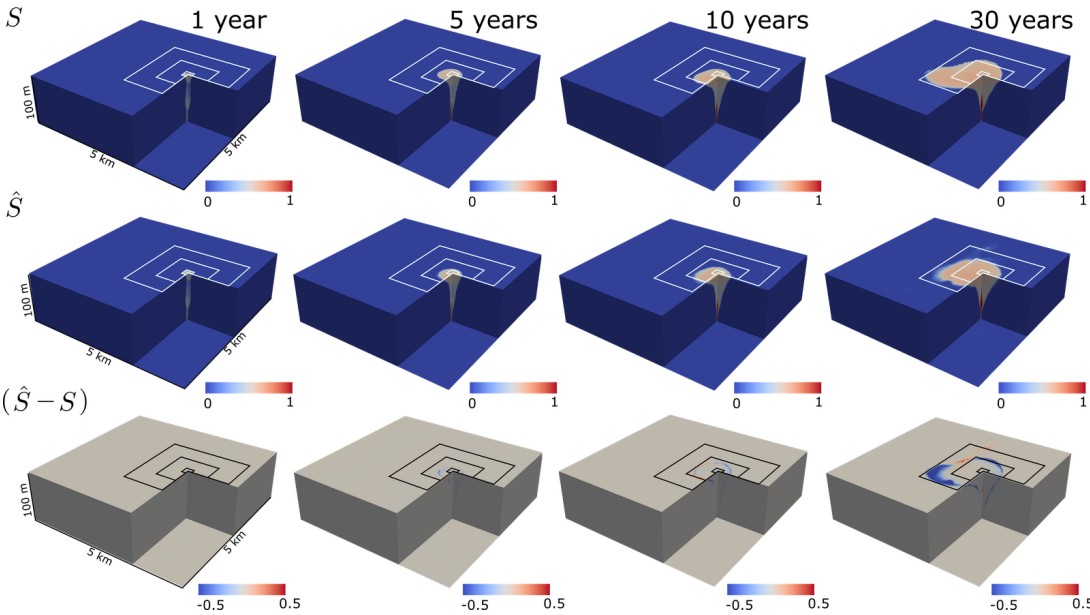

*Figure 11.* Temporal evolution of the gas saturation plume for Well 2. The first and second rows respectively show the gas saturation ($S_g$) fields predicted by APT and the ECLIPSE simulations (HF) for Well 2 at five different times. The third row shows the saturation error ($\delta s_g$) between APT and HF.

*Table 14.* Quantitative comparison on the CarBench's full test set across various neural surrogate models. All models are evaluated with batch size 1 on an NVIDIA A100-SXM4-80GB GPU under identical inference settings. Following the measurement recommended in (Elrefaie et al., 2025), uncertainties are rounded to two significant digits and central values are rounded to the same decimal place; precision is standardized per column using the largest uncertainty in that column ($R_{\text{test}}^2$: $10^{-2}$%, Rel L2: $10^{-4}$). Modified from (Elrefaie et al., 2025).

| Model | Parameters (M) | Peak Memory (GB) | Mean Latency (ms) | $R_{\text{test}}^2$ (%) | Rel L2 |
|---|---|---|---|---|---|
| PointNet[Qi et al., 2017] | 1.67 | 0.29 | 1.54 | $76.39 \pm 0.27$ | $0.3803 \pm 0.0020$ |
| NeuralOperator[Li et al., 2021] | 2.10 | 0.04 | 2.14 | $85.03 \pm 0.26$ | $0.3016 \pm 0.0019$ |
| PointMAE[Pang et al., 2022] | 1.67 | 0.39 | 3.11 | $87.91 \pm 0.23$ | $0.2713 \pm 0.0016$ |
| PointNetLarge[Qi et al., 2017] | 32.58 | 1.50 | 8.29 | $90.25 \pm 0.22$ | $0.2436 \pm 0.0013$ |
| RegDGCNN[Elrefaie et al., 2024a] | 1.44 | 27.11 | 231.98 | $93.27 \pm 0.33$ | $0.2006 \pm 0.0016$ |
| PointTransformer[Zhao et al., 2021] | 3.05 | 6.65 | 95.68 | $93.59 \pm 0.54$ | $0.1909 \pm 0.0024$ |
| TripNet[Chen et al., 2025] | 24.10 | 2.94 | 15.49 | $95.90 \pm 0.67$ | $0.1608 \pm 0.0024$ |
| Transolver++[Wu et al., 2025] | 1.81 | 1.30 | 28.47 | $95.43 \pm 0.66$ | $0.1573 \pm 0.0023$ |
| Transolver[Wu et al., 2024a] | 2.47 | 1.51 | 29.84 | $95.77 \pm 0.67$ | $0.1503 \pm 0.0024$ |
| TransolverLarge[Wu et al., 2024a] | 7.58 | 1.68 | 28.41 | $95.95 \pm 0.69$ | $0.1457 \pm 0.0025$ |
| AB-UPT[Alkin et al., 2025] | 6.01 | 0.27 | 30.65 | $96.75 \pm 0.19$ | $0.1358 \pm 0.0024$ |
| APT (ours) | 6.70 | 0.51 | 35.00 | $96.00 \pm 0.18$ | $0.1535 \pm 0.0022$ |

blocks in the decoder (each containing paired self-attention and cross-attention mechanisms). We found that using only 1024 supernodes for this dataset suffices to achieve good performance. The input projection layer maps $d_{in} = 3$ features (corresponding to 3D surface coordinates) to the latent space. The model is trained for 200 epochs using the AdamW optimizer (learning rate $10^{-4}$, weight decay $10^{-4}$) and a relative $L_p$ loss function (Equation 6).

**Baselines.** We compare against eleven models spanning neural operators (NeuralOperator/FNO (Li et al., 2021)), point cloud architectures (PointNet (Qi et al., 2017), PointNetLarge (Qi et al., 2017), PointMAE (Pang et al., 2022), RegDGCNN (Elrefaie et al., 2024a), PointTransformer (Zhao et al., 2021)), implicit representations (TripNet (Chen et al., 2025)), and physics transformers (Transolver (Wu et al., 2024a), Transolver++ (Wu et al., 2025), AB-UPT (Alkin et al., 2025)). All baseline results are from Elrefaie et al. (2025) under identical training and evaluation protocols.

**Additional Results.** Table 14 provides a comprehensive quantitative comparison of eleven state-of-the-art machine learning models evaluated on the kinematic surface pressure fields of the CarBench test dataset. Although APT does not achieve the best performance, it ranks second among all models in terms of mean $R^2$ scores. We note that both metrics exhibit larger standard deviations compared to other models. This is attributed to differences in our experimental setup: we reproduced the testing protocol based on the paper description rather than the official codebase, which had not been released at the time of our experiments. Additionally, the sampling script was not provided, so we presampled the data using our own random seed while following the described training protocol. We also observed three missing cases in the online dataset. These factors might contribute to edge cases that result in more spread in our results. We made our best effort to match the described protocol to ensure a fair comparison.

Figures 12 and 13 provide a qualitative comparison of surface pressure predictions for the representative design `E_S_WW_WM_648` from the unseen test set, corresponding to an Estateback configuration with a smooth underbody and open wheels. Visually, APT produces the smoothest and most accurate pressure field among all models, with minimal error concentrated only at geometric discontinuities such as wheel wells and panel edges. This observation is consistent with the quantitative super-resolution results reported in Table 5, which demonstrate APT's superior generalization capability. Note that the baseline results are reproduced from (Elrefaie et al., 2025), whereas APT's visualization uses the same colorbar limits but with a standard jet colormap; we attempted to match the visualization style as closely as possible.

## C.2. Created Datasets

The following datasets were generated specifically for this work to evaluate APT on adaptive meshes and out-of-distribution generalization scenarios.

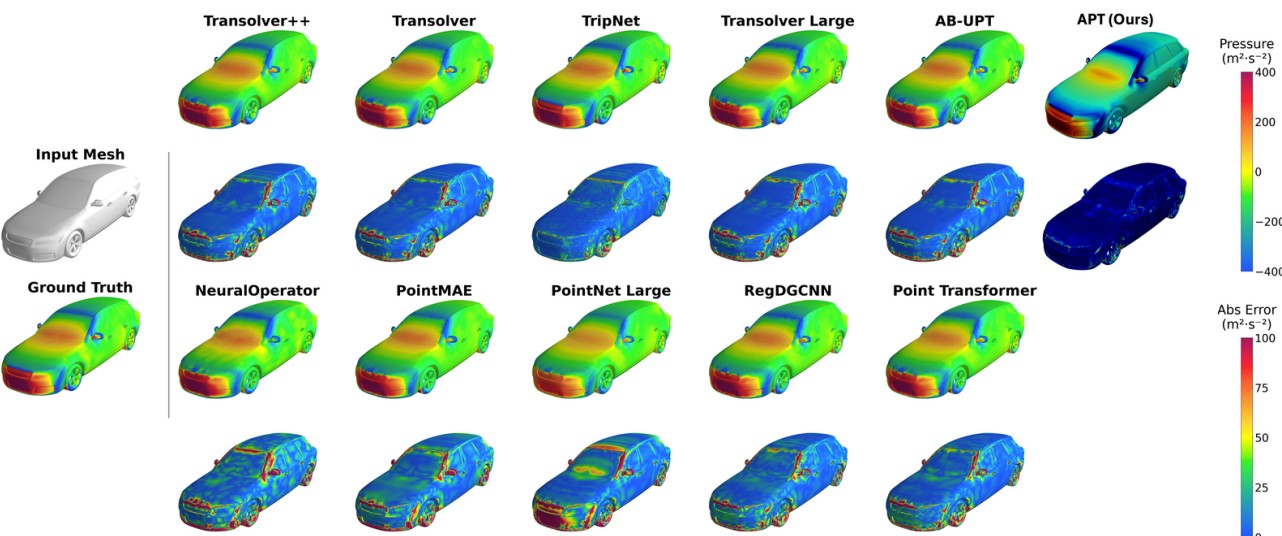

*Figure 12.* Qualitative comparison of surface pressure predictions for design E_S_WW_WM_648 from the unseen test set of the DrivAerNet++ dataset (isometric view). Each row compares predicted pressure fields (top) and absolute error maps (bottom) against the CFD ground truth. APT produces visually smooth predictions across the surface. Note that the baseline results are reproduced from (Elrefaie et al., 2025), whereas APT's result is generated using the same colorbar limits but with a jet colormap; we attempted to match the visualization as closely as possible. Modified from (Elrefaie et al., 2025).

### C.2.1. ADAPTIVE MESH AQUIFER THERMAL ENERGY STORAGE

Aquifer Thermal Energy Storage (ATES) is a shallow geothermal technology that operates on seasonal capture, storage and re-use of excess thermal energy (heat / cool) in shallow subsurface aquifers, providing heating and cooling to the built environment. (Jackson et al., 2024). ATES operates on storing thermal energy seasonally by injecting/extracting groundwater through warm–cold well doublets. Accurate design and operation require numerical simulation of coupled groundwater flow and heat transport, but high spatial resolution is needed near wells and at advancing thermal fronts, making conventional fixed-grid simulations computationally expensive for multi-scenario studies (Regnier et al., 2022)(Jackson et al., 2024).

**Numerical simulator and Dynamic Mesh Optimization (DMO)**  We generate the ATES dataset using the open-source *Imperial College Finite Element Reservoir Simulator* (IC-FERST), which solves flow and heat transport on unstructured tetrahedral meshes with a mesh-adaptive DCVFEM discretisation (Regnier et al., 2022).

IC-FERST employs Dynamic Mesh Optimization (DMO) to reduce computational cost while maintaining user-specified accuracy. DMO is an *anisotropic hr*-adaptivity procedure that refines, coarsens, and repositions the unstructured tetrahedral mesh to minimise a mesh-quality functional derived from an interpolation-error estimate based on the Hessian of selected solution fields and a user-prescribed target precision (Regnier et al., 2022; Kampitsis et al., 2020; Bahlali et al., 2022). The Optimization proceeds via local mesh operations including node movement, element splitting by node insertion, coarsening by node deletion, and face-edge swapping, and the mesh is updated only when these operations reduce the estimated error subject to constraints such as minimum/maximum edge length, maximum aspect ratio and maximum node count (Regnier et al., 2022; Kampitsis et al., 2020). After each mesh update, solution fields are transferred from the old mesh to the new mesh using a supermesh-based remapping with conservative, bounded interpolation (Regnier et al., 2022). In our ATES simulations, the mesh is optimised throughout the transient cycle using pressure and temperature as adaptation fields, yielding trajectories with *time-varying* mesh topology, with node density concentrated near wells and along evolving thermal plumes (Regnier et al., 2022).

**Governing Equations**  The dataset is generated by solving single-phase groundwater flow coupled with heat transport in a porous medium. Following prior ATES modelling with IC-FERST, we assume small temperature-induced density variations

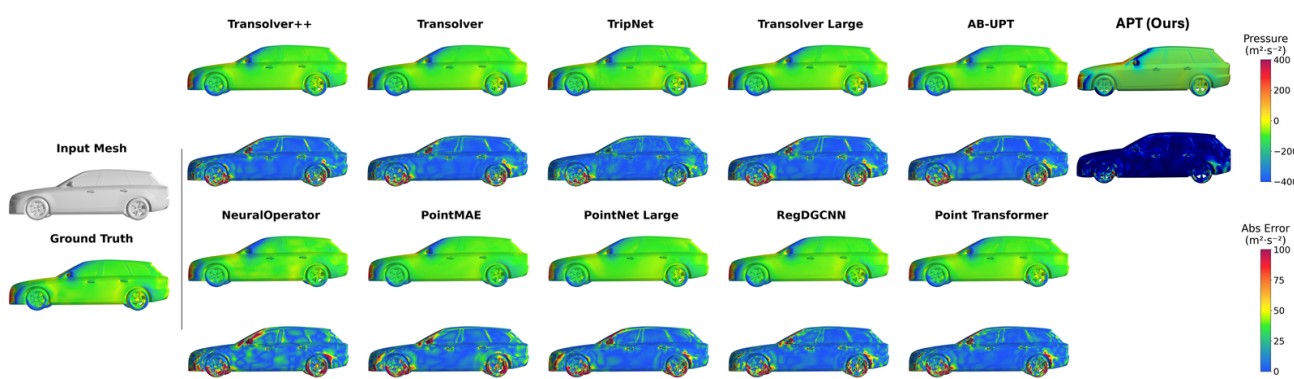

*Figure 13.* Qualitative comparison of surface pressure predictions for design E_S_WW_WM_648 from the unseen test set of the DrivAerNet++ dataset (side view). Each row compares predicted pressure fields (top) and absolute error maps (bottom) against the CFD ground truth. APT produces visually smooth predictions across the surface. Note that the baseline results are reproduced from (Elrefaie et al., 2025), whereas APT's result is generated using the same colorbar limits but with a jet colormap; we attempted to match the visualization as closely as possible. Modified from (Elrefaie et al., 2025).

and neglect thermal dispersion. Flow is described by Darcy's law

$$\mathbf{u} = -\frac{\mathbf{K}}{\mu}\left(\nabla p - \rho_f \mathbf{g}\right), \tag{18}$$

together with mass conservation (constant $\rho_f$)

$$\nabla \cdot \mathbf{u} = \frac{s_c}{\rho_f}, \tag{19}$$

where $\mathbf{u}$ is Darcy velocity, $p$ pressure, $\mathbf{K}$ permeability tensor, $\mu$ viscosity, $\rho_f$ fluid density, $\mathbf{g}$ gravity, and $s_c$ is a source/sink term coupling the reservoir and wells.

Heat transport is modelled by an advection–diffusion equation in thermal equilibrium between fluid and porous medium,

$$\xi\frac{\partial T}{\partial t} + \nabla \cdot \left(\rho_f C_{P,f}\mathbf{u}T - \kappa\nabla T\right) = s_t, \tag{20}$$

with effective volumetric heat capacity and conductivity

$$\xi = (1-\phi)\rho_p C_{P,p} + \phi\rho_f C_{P,f}, \qquad \kappa = (1-\phi)\kappa_p + \phi\kappa_f, \tag{21}$$

where $T$ is temperature, $\phi$ porosity, and subscripts $p, f$ denote porous medium and fluid, respectively; $s_t$ represents well–reservoir heat exchange.

**Dataset generation and splits.**   We simulate cyclic seasonal operation over 10 years with 240 timesteps (2 timesteps per month). Each annual cycle consists of winter operation (cold well injection / warm well extraction), transitional resting periods (no injection/extraction in both well), summer operation (warm well injection / cold well extraction), and another resting period. The synthetic dataset contains 840 scenarios with varied well configurations and vertical heterogeneity. Key scenario variables are sampled within the following ranges: well spacing $[12.5, 500]$ m; well depth (warm/cold) $[-150, -60]$ m; screen length $[10, 100]$ m; screen separation $[0, 90]$ m; and vertical permeability $K_z \in [1, 1000]$ (units as in the simulator setup). We split scenarios into 80% training, 10% validation, and 10% test.

**Learning task (direct one-step).**   Our benchmark focuses on predicting temperature as a node attribute on the adaptive mesh. In contrast to autoregressive rollout, we use a *direct* one-step setting: given the initial condition at $t{=}0$ (and static properties / controls) plus a query time $t$, the model predicts $T(\mathbf{x}, t)$ on the target adaptive mesh without iterative stepping through intermediate states. This evaluates the learned solution operator while avoiding temporal error accumulation, and directly reflects the challenges posed by DMO-driven changing node distributions.

**Fixed-grid baselines (U-Net/FNO).**   To benchmark against standard operator baselines that require structured inputs, we construct a fixed Cartesian grid representation of each ATES state by interpolating the adaptive-mesh solution fields onto a regular voxel grid. For each timestep, the adaptive mesh node coordinates and temperatures are mapped to the fixed grid using pointwise interpolation. The resulting tensor has four input channels, where the first channel is the current temperature field and the remaining channels encode time-invariant geological properties and/or operational controls used by the baseline models. This produces paired training samples $(\mathbf{x}_t, \mathbf{y}_t)$ on the fixed grid, where $\mathbf{x}_t \in \mathbb{R}^{4 \times N_x \times N_y \times N_z}$ and $\mathbf{y}_t \in \mathbb{R}^{1 \times N_x \times N_y \times N_z}$ represents the next-step temperature.

We adopt a 3D U-Net with residual bottleneck blocks. The network follows a four-level encoder-decoder hierarchy with channel widths $C \to 2C \to 4C \to 8C$ and skip connections, and applies $n_{\mathrm{res}}$ residual blocks at the bottleneck. The best model uses $C{=}48$ and $n_{\mathrm{res}}{=}8$, with residual scaling 0.1 and a zero-initialized output head. We also implement a 3D Fourier Neural Operator (FNO) (Li et al., 2021) as a baseline model. The FNO uses 4 input channels, 1 output channel, a hidden width of 48, and 4 Fourier layers with mode truncations of $(16, 16, 16)$ along the three spatial dimensions.

**Mesh-native baselines (UPT, Transolver, MINO).**   For comparison against mesh-native operators we evaluate three transformer-based baselines on ATES under the same direct one-step setting as APT (initial condition + query time $t \to T(\mathbf{x}, t)$ on the target adaptive mesh). UPT (Alkin et al., 2024) uses a supernode encoder with $\dim{=}192$, 4 encoder / 4 approximator / 4 decoder layers, 1024 supernodes, and 128 latent tokens. Transolver (Wu et al., 2024a) uses $n_{\mathrm{hidden}}{=}640$, 8 layers, 8 heads, and 64 slices to match APT's representational capacity on the larger ATES domain. MINO (Shi et al., 2025) could not be evaluated: its latent tensor $[L_{\dim} \times N_{\mathrm{node}}]$ exceeds 80 GB on an A100 GPU at the ATES sampling of 8,192 nodes per snapshot, demonstrating the practical memory limitation of architectures whose latent state scales with mesh size. Both UPT and Transolver are trained for 50 epochs on 4,096-node samples with batch size 16, AdamW optimizer (learning rate $10^{-3}$, weight decay $10^{-5}$), and a onecycle schedule. APT reaches $R^2{=}99.0\%$ (relative $L_2$ error 0.011), outperforming Transolver ($R^2{=}98.6\%$, error 0.0237) and UPT ($R^2{=}96.0\%$, error 0.0401) on the same test set.

**Inference and evaluation on adaptive meshes.**   At test time, U-Net/FNO predictions are rolled out autoregressively on the fixed grid: starting from the initial fixed-grid state, the predicted temperature at step $t$ is fed back as the temperature input channel for step $t{+}1$, while the remaining input channels remain unchanged. To compare fairly against APT (which predicts directly on the adaptive meshes), we evaluate the fixed-grid rollouts on the original adaptive-mesh nodes by interpolating (sampling) the fixed-grid prediction back to the adaptive coordinates at each timestep using trilinear sampling. Metrics (e.g., $R^2$) are then computed on the adaptive-mesh nodes in physical temperature units.

**APT implementation.**   The APT architecture for ATES follows the same design principles as the Nested LGR implementation (Section C.1.3), with modifications to accommodate the dataset's characteristics. At each training step, input grids are randomly sampled with a fixed size of 8,192 cells. Key architectural differences include: (1) the conditioner processes 7 input field features (compared to 3 in Nested LGR) with a correspondingly adjusted input projection, and (2) the perceiver module compresses to 512 latent tokens (half of Nested LGR's 1,024 tokens). The model uses a hidden dimension of 192, a conditioning dimension of 768, and an MLP expansion ratio of 4, totaling 17.5M parameters.

### C.2.2. WASTEWATER INJECTION

In industries such as oil and gas, geothermal, and chemical engineering, large volumes of wastewater are generated and are difficult to treat at the surface. These fluids are commonly injected into deep subsurface formations, such as deep saline aquifers or depleted hydrocarbon reservoirs, to achieve long-term isolation. Under such conditions, the associated increase in pore pressure may trigger fault slip, potentially inducing seismic events. Therefore, effective estimation of the reservoir pressure field prior to wastewater injection is of critical importance for safe and reliable operation.

**Dataset Generation.**   We utilize the numerical simulator ECLIPSE (Schlumberger, 2014) to develop the wastewater injection dataset. A vertical injection well with a radius of 0.1 m delivers wastewater at a constant rate into a radially symmetric system $x(r, z)$. The well completion may span the full reservoir thickness or be restricted to a selected depth interval. Here, the reservoir thickness ranges from 125 to 500 m, with no-flow boundary conditions imposed at the top and bottom. The radius of the reservoir is set as 100,000 m, resolved with 200 gradually coarsened grid cells. Simulated pressure buildup ($dP$) fields at 24 time snapshots are used to train the model, spanning prediction horizons from short term (30 days) to long term (30 years). For each snapshot, the field variable include the horizontal permeability map ($k_r$), vertical permeability map ($k_z$), porosity map ($\phi$), and the injection perforation map ($perf$). All simulation scenarios follow the

heterogeneous setup introduced in (Wen et al., 2022). The scalar inputs include injection rate and initial field pressure, which are uniformly sampled from [5434, 27170] STB/day and [250, 450] bar to increase the data distribution complexity.

Overall, we generated 4000 cases, of which 3000 are used for training, 500 for validation, and 500 for testing. It should be noted that in this dataset we consider a structure-aware out-of-distribution generalization setting. Specifically, the training set spans only a subset of permeability field configurations, namely continuous Gaussian, continuous von Karman, and discontinuous Gaussian fields. The test set, however, consists solely of discontinuous von Karman permeability fields. Figure 14 shows this distribution shift: three randomly selected permeability maps from the training set (continuous Gaussian, continuous von Kármán, and discontinuous Gaussian) are shown alongside a representative discontinuous von Kármán case from the out-of-distribution test set.

**Training distribution**           **OOD test distribution**

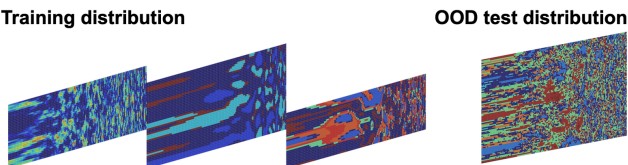

*Figure 14.* OOD evaluation setup. The model is trained on a mixture of continuous/discontinuous Gaussian and continuous von Karman fields, but tested on a held-out class of **discontinuous von Karman** fields, requiring generalization to unseen geological structures.

*Table 15.* OOD Generalization Results. Quantitative comparison on the held-out discontinuous von Karman test set. The **APT (fused)** variant achieves the highest accuracy, outperforming both pure spectral baselines (FNO, TFNO) and single-branch ablations.

| Model | Test (OOD) | | Train | |
|---|---|---|---|---|
| | $R^2$ | Rel. Err | $R^2$ | Rel. Err |
| FNO | 0.9052 | 0.0376 | 0.9306 | 0.0297 |
| TFNO | 0.9079 | 0.0362 | 0.9241 | 0.0291 |
| *APT variants (ablation)* | | | | |
| APT (local only) | 0.8629 | 0.0404 | 0.8707 | 0.0354 |
| APT (global only) | 0.8917 | 0.0355 | 0.8970 | 0.0341 |
| **APT (fused)** | **0.9107** | **0.0328** | **0.9351** | **0.0280** |

### C.2.3. NESTED GRID CHANNELIZED RESERVOIR CO$_2$ STORAGE

**Dataset Configuration.** To rigorously evaluate the model's capability in learning from multiple large-scale datasets, we create the Channelized Reservoir dataset via following the basic simulation setup of LGR dataset, detailed in Section C.1.3. Similarly, the simulation encompasses variable injection scenarios ranging from one to four active wells, testing the surrogate model's adaptability to varying source terms. However, we design two important variations when generating the simulation data, namely geomodel parameterization and mesh topology.

**Geomodel Parameterization.** The dataset adapts a geologically realistic, channelized facies architecture. As illustrated in Figure 15, the permeability field is characterized by a strongly bimodal distribution. This binary system comprises a background mudstone facies (dominant mode at $\ln(k) \approx 3.5$, ~33 mD) and high-permeability channel sand bodies (secondary mode at $\ln(k) \approx 6.5$, ~665 mD). This bimodal structure introduces sharp permeability contrasts and discontinuous channel geometries, thereby imposing significant complexity on the flow dynamics and presenting a more challenging learning task for the neural operator compared to smooth Gaussian media.

**Nested Grid Topology.** We employ a modified hierarchical LGR structure to capture the multi-scale flux interactions induced by the channelized permeability. The refinement strategy, detailed in Table 16, differs substantially from the baseline LGR dataset. We prioritize finer discretization in the near-well regions (LGR3–LGR5) with a $20 \times 20 \times 2$ m cell resolution to resolve high-velocity flow and pressure gradients. This results in highly variable mesh topologies across samples, with total grid counts scaling between 0.5 and 0.8 million cells depending on the active refinement levels.

*Table 16.* Mesh refinement hierarchy for the channelized reservoir dataset. The configuration emphasizes high-resolution local grids (LGR3–5) to resolve sharp flow gradients within channel facies.

| Refinement Level | Grid Dimensions | Resolution (m) | Cell Count |
|---|---|---|---|
| Global | $100 \times 100 \times 5$ | $1600 \times 1600 \times 20$ | 49,200 |
| LGR1 | $40 \times 64 \times 50$ | $400 \times 400 \times 2$ | 108,000 |
| LGR2 | $40 \times 40 \times 50$ | $200 \times 200 \times 2$ | 74,600 |
| LGR3–LGR5 | $60 \times 60 \times 50$ | $20 \times 20 \times 2$ | 180,000 (each) |

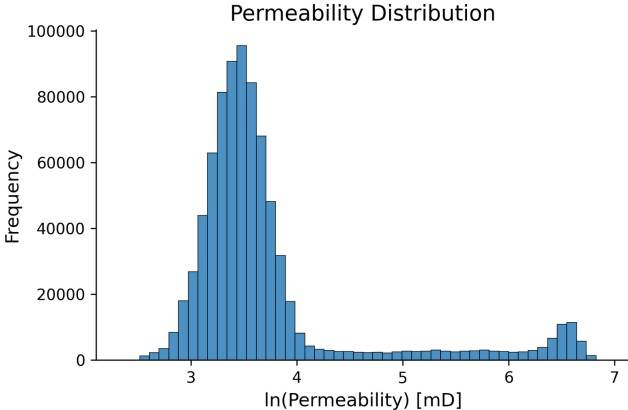

*Figure 15.* Permeability distribution of the channelized reservoir model. The histogram reveals a distinct bimodal signature: a low-permeability background facies ($\ln(k) \approx 3.5$) and high-permeability channel sands ($\ln(k) \approx 6.5$). This sharp contrast approximates binary geological media, testing the model's ability to preserve discontinuous interfaces.

**APT implementation.** To facilitate the multi-dataset training, we use the same APT architecture and training protocol, as described in Section C.1.3.

# D. Generalization to Standard PDE Benchmarks

To assess the architectural breadth of APT beyond the subsurface datasets, we evaluate it on three widely used PDE benchmarks: Airfoil (compressible flow), Darcy (linear porous-medium flow), and Elasticity (linear solid mechanics). Following the protocol of Wu et al. (2024a), we report the mean relative $L_2$ error on the canonical test splits for the three mesh-native transformer baselines used elsewhere in this work, together with APT.

*Table 17.* Relative $L_2$ error on three homogeneous-domain PDE benchmarks. Lower is better. APT is consistently the second-best model, behind Transolver.

| Dataset | Transolver | MINO | UPT | APT |
|---|---|---|---|---|
| Airfoil | **0.00553** | 0.04052 | 0.00865 | 0.00625 |
| Darcy | **0.00592** | 0.00702 | 0.01225 | 0.00713 |
| Elasticity | **0.00682** | 0.03032 | 0.06972 | 0.00960 |

APT is consistently the second-best model across the two out of three benchmarks, trailing Transolver by a small margin on each dataset. This extra comparison on general PDE benchmarks, supports an architecturally physics-agnostic interpretation of APT. The model can be trained on standard PDE problems and remains competitive with specialized operators. However, we find that APT's empirical advantage is most pronounced on the heterogeneous, multi-scale subsurface settings.

