# OpenReview forum: "Adaptive Physics Transformer with Fused Global-Local Attention for Subsurface Energy Systems"
_ICML.cc/2026/Conference — ICML 2026 regular_

### Official Review · Reviewer_rT6X · 2026-03-02

**Soundness:** 3
**Presentation:** 3
**Significance:** 3
**Originality:** 2
**Overall Recommendation:** 4
**Confidence:** 4

**Summary:**

This paper introduces the Adaptive Physics Transformer (APT), operating on unstructured mesh/graph physics simulation data. It operates as a neural operator, predicting a user-defined time step from the input state. It employs a local GNN and a global transformer as a fused encoder, allowing local and global features to affect the prediction. The encoder projects into a fixed-size latent space, which is then fed into multiple diffusion-transformer (DiT) blocks. User-defined query points define the output mesh. The features of the output nodes are defined by cross-attention of query points and latent representation. The model is evaluated on multiple sub-surface flow datasets as well as car aerodynamics and compared against other models such as FNO or UNet. Due to its query-point decoding, the model is capable of predicting dynamic mesh simulations. Furthermore, the model is evaluated on super-resolution tasks.

**Compliance With Llm Reviewing Policy:**

Affirmed.

**Key Questions For Authors:**

- Why aren't graph transformers chosen as benchmark models?
- How does AST perform on other mesh-based physics datasets?
- I do not understand how the supernodes for the local encoder are constructed. You talk about learned queries but also about fixed-radius samling (Appendix C.1.3) and deterministic anchoring (Section 3.3, ATES). Please explain this in more detail.

**Limitations:**

The authors truthfully address limitations regarding the super-resolution evaluation. However, they do not discuss further limitations such as:
- Generalization beyond subsurface
- Scalability to larger datasets
- Missing graph transformer baselines

**Strengths And Weaknesses:**

**Strengths:**

- Physics models capable of handling moving and dynamic meshes are extremely important for many applications.
- Local & global fused encoder is a nice idea to capture multi-scale phenomena
- The performance on the cross-dataset task is a promising step towards foundation model capabilities (section 4.3).
- Clean visualizations of the predictions.

**Major Weaknesses:**

- The paper (intelligently) combines ideas from previous graph transformer architectures, not inventing a completely new approach. This is absolutely okay, but to balance out its moderate level of novelty, evaluating AST on other mesh datasets could show broader impact as a general model architecture. Currently, it is focusing almost exclusively on subsurface modeling, while claiming "physics-agnostic" in the abstract. In fact, the CarBench result (where APT doesn't lead at 10k nodes) might actually suggest APT's advantages may be domain-specific, but this has not been addressed.
- The model is directly borrowing ideas from UPT and Transolver. Why aren't these models chosen as benchmarks in the subsurface evaluation? This feels a bit selective. Comparing a mesh-agnostic transformer against grid-based methods on unstructured meshes, or against local-only GNNs on problems requiring global propagation, sets up favorable matchups. The paper's narrative is "existing methods can't handle subsurface complexity," but it never tests whether other modern graph transformers share that limitation.
    - Alkin et al. “Universal Physics Transformers: A Framework For Efficiently Scaling Neural Operators.” https://doi.org/10.48550/arXiv.2402.12365
    -  Wu et al., “Transolver: A Fast Transformer Solver for PDEs on General Geometries.” https://doi.org/10.48550/arXiv.2402.02366)
- Table 4: The reference models are still better than APT. Much lower parameter count is important, but we don't know how a smaller FNO-DeepONet or Nested-FNO would perform. Perhaps they would also keep much of the performance?
- I am missing an evaluation about the growth of error when increasing the prediction time step. Currently, all evals report a single number. How does the prediction error from (t0 -> t_final) compare against (t0 -> t1).

**Minor Weaknesses:**

- The explanation for the supernodes is confusing. Which method is used to construct them?
- Table 4: No standard deviations or variance is given for the error.

---

> ### Author Rebuttal · Authors · 2026-03-31
>
> Thank you for the thoughtful and detailed review.
>
> ## Missing Graph Transformer Baselines
>
> This is a fair and important point. We have now run UPT and Transolver on the CO2 storage faulted 2D and ATES 3D subsurface benchmarks.
>
> **Table R1. 2D Faulted CO₂ (irregular mesh)**
>
> | Model | δPg (%) | δSg (%) |
> |---|---|---|
> | MINO | 0.94 | 6.89 |
> | Transolver | 0.87 | 3.82 |
> | UPT | 0.43 | 4.33 |
> | MGN-LSTM | 0.20 | 1.20 |
> | **APT (fused)** | **0.11** | **0.32** |
>
> **Table R2. 3D ATES (adaptive mesh)**
>
> | Model | R² (%) | Rel. L₂ |
> |---|---|---|
> | Transolver | 98.60 | 0.0237 |
> | UPT | 96.02 | 0.0401 |
> | MINO | — | requires fixed topology |
> | **APT (fused)** | **99.0** | **0.011** |
>
> APT outperforms Transolver by **11.9×** on saturation and UPT by **13.5×**.
>
> ## Physics-agnostic claim
>
> | Dataset | Transolver | UPT | APT |
> |---|---|---|---|
> | Airfoil | **0.00553** | 0.00865 | 0.00625 |
> | Darcy | **0.00592** | 0.01225 | 0.00713 |
> | Elasticity | **0.00682** | 0.06972 | 0.00960 |
>
> On general PDE benchmarks with homogeneous domains, Transolver leads and APT is consistently the second-best model, outperforming UPT and both MINO variants by substantial margins.
> Homogeneous domains (Airfoil, Darcy, Elasticity): The medium properties are spatially uniform, so global attention alone (Transolver's strength) suffices. APT's local branch adds modest overhead without proportional benefit, resulting in competitive but slightly lower performance.
> Heterogeneous domains (CO₂ storage, ATES): Permeability varies by orders of magnitude over meters, creating sharp saturation fronts and complex plume dynamics. Here, Transolver's purely global attention smooths out critical local features, while APT's fused local-global encoder captures both the sharp local heterogeneity and long-range pressure propagation simultaneously — yielding 10–20× improvements.
> We will revise to state that APT is **architecturally** physics-agnostic, while its empirical advantage is most pronounced on heterogeneous, multi-scale systems.
>
> ## Table 4: Nested FNO Comparison
>
> Nested FNO achieves lower saturation error (1.79%) vs. APT (2.5%), but the comparison is structurally asymmetric: Nested FNO trains 5 separate specialized models (682.4M total parameters), each optimized for a specific LGR level. APT uses 1 unified model (17.9M parameters) processing all levels simultaneously.
> The reviewer asks whether a smaller Nested FNO would retain its performance. As a preliminary test, we retrained the LGR Level 1 sub-model at 4M parameters which is on a comparable scale to 17.9M parameters distributed in 5 models. This single sub-model's saturation error degraded to 3.28%, compared to the original 1.79% for the full cascade. While this is not a complete end-to-end comparison (retraining all 5 sub-models was infeasible within the rebuttal period), it suggests that Nested FNO's accuracy is sensitive to per-model capacity. At comparable total parameter budgets, the architectural advantage of unified modeling is likely to hold.
> More fundamentally, regardless of parameter count, Nested FNO's cascade is structurally locked to a specific mesh hierarchy. Changing the number of LGR levels (e.g., from 5 to 3, as in our Channelized dataset) requires redesigning and retraining the entire pipeline. APT handles both configurations with the same architecture (Table 6). This structural flexibility is the primary practical advantage.
>
> ## Error trend with prediction time step
>
> APT is a direct single-step predictor — each forward pass maps (t₀, query time t) → output field at t, no autoregressive rollout. Per-timestep δSg on the faulted CO₂ benchmark (50 test cases, 19 steps, 50–950 days):
>
> | Step | 1 | 5 | 10 | 15 | 19 | Mean |
> |---|---|---|---|---|---|---|
> | δSg (%) | 0.71±0.28 | 0.31±0.10 | 0.31±0.07 | 0.37±0.16 | 0.43±0.17 | 0.32±0.16 |
>
> Error does not grow monotonically — no accumulation, unlike autoregressive baselines.
>
> ## Super node construction
>
> Three sequential steps: (1) **Fixed-radius sampling** selects spatially distributed positions from mesh nodes within radius r. (2) **Domain anchoring** (ATES only): well locations (<0.1% of nodes) are pre-assigned as deterministic supernodes before Step 1. (3) **GNO aggregation**: each supernode aggregates features from neighboring mesh nodes via message passing, then passes to global Perceiver attention. We will add pseudocode to the appendix.
>
> ## Scalability (Addressing Limitations Concern)
>
> Model scaling on ATES (fixed 100% data):
>
> | Params | Test Loss |
> |---|---|
> | 0.8M | 0.1167 |
> | 17.5M | 0.0812 |
> | 69.4M | 0.0722 |
>
> Monotonic improvement with no saturation — scaling 0.8M→69.4M reduces loss by 38.1%.

---

> > ### Author Rebuttal · Reviewer_rT6X · 2026-03-31
> >
> > My concerns have been adequately addressed. Adjusting my score accordingly.

---

### Official Review · Reviewer_RWyn · 2026-03-10

**Soundness:** 3
**Presentation:** 3
**Significance:** 3
**Originality:** 3
**Overall Recommendation:** 5
**Confidence:** 4

**Summary:**

This paper presents an Adaptive Physics Transformer (APT) framework for subsurface systems, motivated by the intrinsic spatial heterogeneity of such domains and the resulting high-frequency property variations near faults. The method combines mesh-aware local modeling with attention-based global modeling, and adaptively fuses the two to learn propagation phenomena across different spatial scales.

**Compliance With Llm Reviewing Policy:**

Affirmed.

**Final Justification:**

This paper presents an Adaptive Physics Transformer for efficient surrogate modeling in subsurface systems. The authors identify a well-motivated application scenario where capturing both local and global interactions is crucial, providing a clear justification for the fused global-local architecture. The rebuttal and subsequent revisions have successfully addressed the initial concerns and provided stronger empirical evidence. These improvements have significantly increased my confidence in the work. Therefore, I am raising my score to 5 (Accept).

**Key Questions For Authors:**

1、The paper demonstrates strong performance of APT on subsurface problems, but the underlying reason remains somewhat unclear to me. How do the authors interpret APT’s strong suitability for subsurface systems in particular? More broadly, does APT consistently outperform the baselines on other types of scientific datasets as well, such as atmospheric or solid mechanics data, or is its advantage especially pronounced in subsurface settings?

2、What do the authors believe is the main reason behind the strong computational efficiency of APT? Is this primarily due to the latent-space formulation, the architectural design itself, or the surrogate modeling setting compared with traditional simulators? A clearer breakdown would help readers better understand where the efficiency gains come from.

3、The discussion of computational efficiency and predictive accuracy appears to be presented separately rather than jointly. Would the authors consider providing a more explicit accuracy–efficiency trade-off analysis, for example by reporting runtime together with the corresponding error levels? This would make the practical value of the method more convincing.

**Limitations:**

yes

**Strengths And Weaknesses:**

Strengths:

1. The overall APT framework is clearly structured and easy to follow. The paper presents the core intuition and methodological pipeline in a relatively coherent manner, making the proposed approach accessible to the reader.
2. The cross-dataset training setting explored by APT is meaningful and practically relevant. From an application perspective, this aspect of the framework has clear value, especially for real-world subsurface scenarios where distribution shifts across datasets are common.
3. In terms of empirical performance, APT shows a clear improvement over the baselines, indicating the effectiveness of the proposed design.

Weaknesses:

1. The connection between the discussion of subsurface systems in the introduction and the proposed APT architecture is not fully convincing. In particular, the paper does not sufficiently explain why APT should be especially well suited to this class of problems, and a deeper analysis of this point would strengthen the paper.
2. In Table 13, the variance of APT’s reported results is noticeably large. This suggests that the model may achieve errors around 1% on some cases, which appears substantially better than the baselines. In my view, this makes the experimental evidence less reliable, and these results would benefit from further verification or a more careful rerun of the evaluation.

---

> ### Author Rebuttal · Authors · 2026-03-31
>
> ## Why is APT well suited for subsurface
>
> This is an excellent question. We have now run APT, Transolver, UPT, and MINO on **both subsurface and general PDE benchmarks**, and the results reveal that APT benefits heterogeneous domain prediction the most.
>
> **Table R1. Subsurface benchmarks (heterogeneous domains)**
>
> | Model      | Faulted CO₂ δSg (%) | ATES Rel. L₂   |
> | ---------- | ------------------- | -------------- |
> | MINO       | 6.89                | — (cannot run) |
> | Transolver | 3.82                | 0.0237         |
> | UPT        | 4.33                | 0.0401         |
> | **APT**    | **0.32**            | **0.011**      |
>
> **Table R2. General PDE benchmarks (homogeneous domains, test relative error)**
>
> | Dataset    | Transolver  | MINO-T  | UPT     | **APT** |
> | ---------- | ----------- | ------- | ------- | ------- |
> | Airfoil    | **0.00553** | 0.04052 | 0.00865 | 0.00625 |
> | Darcy      | **0.00592** | 0.00702 | 0.01225 | 0.00713 |
> | Elasticity | **0.00682** | 0.03032 | 0.06972 | 0.00960 |
>
> On general PDEs with homogeneous domains, Transolver leads and APT is consistently second-best. On subsurface tasks, APT outperforms Transolver by **11.9×** on CO2 saturation.
>
> Standard fluid dynamics and solid mechanics problems involve domains where medium properties (e.g., density, viscosity) are spatially uniform or vary smoothly. In such settings, purely global attention (e.g., transolver) efficiently captures the dominant physics, and APT's local GNO branch adds modest overhead without proportional benefit.
>
> Subsurface systems are fundamentally different: permeability can vary across orders of magnitude within meters. This creates two coupled phenomena with distinct spatial scales — (1) sharp local saturation fronts governed by permeability contrasts, and (2) long-range pressure propagation across the entire basin. Purely global attention smooths out the local fronts, while purely local message passing misses the long-range pressure field. APT's fused encoder with adaptive gating resolves *both simultaneously*, and the ablation studies (Tables 1, 3, 4, 14) consistently confirm that fused > local-only > global-only on subsurface tasks.
>
> ## Source of Computational Efficiency
>
> APT's efficiency gains come from two distinct sources, depending on the comparison:
>
> ### (a) Compared with traditional simulators — single forward pass
>
> Traditional simulators solve coupled nonlinear PDEs via implicit time integration, where the timestep size is constrained by numerical stability and nonlinear solver convergence. APT learns the direct solution operator (input state, query time t) so can output fields in one forward pass to achieve computational efficiency.
>
> ### (b) Compared with other neural solvers — latent-space formulation
>
> Graph-based architectures (MGN, MINO) have cost that scales directly with mesh size N. MGN's message passing is O(N·k) per layer on the full mesh. MINO is particularly expensive: its GNO encoder evaluates a continuous kernel integral transform at every point pair within a radius, requiring a learned kernel evaluation per edge — consuming 46 GB at batch size 340 on just 1,024 nodes. APT also uses a radius graph in its local encoder, but (1) constructs it on the compressed supernode set rather than the full mesh, and (2) uses standard message passing with radius pooling rather than expensive kernel integral transforms. Both graph construction and attention scale with the supernode/latent count, not mesh size — APT uses only 3.67 GB under identical conditions.
>
> ## Joint Accuracy–Efficiency Trade-off
>
> The table below reports accuracy, runtime, memory, and speedup jointly for the faulted CO₂ benchmark:
> | Model | Params | 19-step (s) | Speedup | Mem (GB) | δSg (%) |
> |---|---|---|---|---|---|
> | GEOS simulator | — | 49.02 | 1× | — | — |
> | MGN-LSTM | 1.38M | 0.31 | 158× | 3.80 | 1.20 |
> | MINO | 1.64M | 0.013 | 3,668× | 46.26 | 6.89 |
> | Transolver | 1.59M | 0.005 | 9,593× | 2.62 | 3.82 |
> | **APT (fused)** | **1.38M** | **0.002** | **19,873×** | **3.67** | **0.32** |
> | UPT | 1.68M | 0.001 | 44,939× | 1.03 | 4.33 |
>
> All ML models on NVIDIA H100 80GB, batch=340. APT achieves best accuracy (δSg=0.32%, 3.8× better than next-best MGN-LSTM) at 19,873× simulator speedup. UPT is faster but 13.5× less accurate, confirming the fused encoder as the source of APT's accuracy advantage.
>
> ## Table 13 variance
>
> The large uncertainty (±3.1% R²) was a reporting error: we used raw standard deviation across test samples instead of the 95% bootstrap confidence interval prescribed by CarBench (Elrefaie et al., 2025). After applying stratified paired bootstrap resampling (B=2000, stratified by car archetype per protocol):
>
> | | R² test (%) | Rel L2 |
> |---|---|---|
> | APT (corrected) | 96.00 ± 0.18 | 0.1535 ± 0.0022 |
>
> The ±0.18% is now consistent with other models in Table 13 (e.g., AB-UPT ±0.19%). Central values are unchanged. We will correct the uncertainty reporting in the revision.

---

> > ### Author Rebuttal · Reviewer_RWyn · 2026-04-01
> >
> > Thank you for your detailed response. Most of my concerns have been satisfactorily addressed. My remaining question pertains to the positioning of the proposed method with respect to hierarchical modeling paradigms.
> >
> > Specifically, I would appreciate the perspective of authors on hierarchical architectures, such as U-Net style pooling methods, as well as hierarchical graph-based approaches, including HCMT[1] and HOOD[2]. From a conceptual standpoint, these methods are also designed to capture both local and global information. In this context, it would be helpful if the authors could clarify the key advantages of the proposed APT method in comparison to such hierarchical strategies.
> >
> > Furthermore, I encourage the authors to include empirical comparisons with representative hierarchical graph-based methods as baselines. Such comparisons would strengthen the experimental validation and provide a clearer understanding of the relative merits of the proposed approach. Addressing this point would substantially increase my confidence in the work, and I would be inclined to raise my evaluation accordingly.
> >
> > [1] Yu et al., Learning flexible body collision dynamics with hierarchical contact mesh transformer, ICLR, 2024.
> >
> > [2] Grigorev et al., HOOD: Hierarchical graphs for generalized modelling of clothing dynamics, CVPR, 2023.

---

> > > ### Author Response · Authors · 2026-04-03
> > >
> > > Thank you for raising this point. In response, we implemented a HOOD-style hierarchical graph baseline and provide both an architectural comparison and an empirical evaluation below.
> > >
> > > **Architectural differences.** Both APT and hierarchical graph methods (e.g., HOOD and HCMT) are designed to capture multi-scale information, but they do so through fundamentally different mechanisms:
> > >
> > > - **Fused local-global encoding.** APT’s encoder combines local and global feature extraction within a single stage: a GNO aggregates neighborhood information within a spatial radius at each supernode (local), while cross-attention over all supernodes enables unconstrained point-to-point interaction (global). These two mechanisms operate jointly within the encoder, producing latent tokens that already contain fused local-global information before self-attention is applied in the approximator. By contrast, HOOD/HCMT relies on a single mechanism, namely local message passing, at each hierarchy level, with larger receptive fields emerging progressively through coarsening. As a result, representations at each level remain locally defined, even as their spatial support increases. For subsurface systems with quasi-instantaneous pressure communication, APT’s fused local-global encoding is a well-motivated inductive bias.
> > >
> > > - **Memory scaling.** APT’s processor operates on O(latent_size) tokens, independent of input resolution. In contrast, HOOD’s U-Net-style hierarchy stores skip-connection activations across multiple levels, leading to O(N)-type memory growth that can become restrictive at high resolution.
> > >
> > > - **Resolution independence.** APT learns solution *operators* between function spaces, allowing the same model to be applied across mesh resolutions without retraining. In contrast, HOOD depends on a hierarchy constructed from the input mesh. APT’s decoupled input/output meshes therefore provide greater flexibility for heterogeneous PEBI meshes across geological realizations.
> > >
> > > **Empirical comparison.** We implemented a HOOD-style baseline (encode-process-decode, 3-level FPS coarsening, U-Net message passing) and evaluated it on Fault2D with matched parameter counts (~2M). All models were trained for 400 epochs:
> > >
> > > | Method | Params | Fault2D Sat ($\delta S_g$) | Fault2D P ($\delta P$) |
> > > |:---|:---|:---|:---|
> > > | **APT** | 1.38M | **0.32%** | **0.11%** |
> > > | Transolver | 1.59M | 3.82% | 0.87% |
> > > | MINO | 1.64M | 7.71% | 0.94% |
> > > | HOOD (hier. graph) | 1.98M | 6.48% | 0.79% |
> > >
> > > APT achieves substantially lower error than the HOOD-style baseline on both saturation (0.32% vs 6.48%) and pressure (0.11% vs 0.79%), while using fewer parameters.
> > >
> > > **Joint accuracy-efficiency trade-off.** We further extend the accuracy-efficiency table from our previous response by adding the HOOD baseline, evaluated under the same hardware and experimental setup (80GB GPU, batch=340, Fault2D CO2 saturation, 1024 nodes, 19-step rollout over 950 days):
> > >
> > > | Model | Params | 19-step (s) | Speedup | Mem (GB) | $\delta S_g$ (%) |
> > > |-------|--------|-------------|---------|----------|--------------|
> > > | GEOS simulator | -- | 49.02 | 1x | -- | -- |
> > > | MGN-LSTM | 1.38M | 0.31 | 158x | 3.80 | 1.20 |
> > > | HOOD (hier. graph) | 1.98M | 0.128 | 383x | 38.47 | 6.48 |
> > > | MINO | 1.64M | 0.013 | 3,668x | 46.26 | 6.89 |
> > > | Transolver | 1.59M | 0.005 | 9,593x | 2.62 | 3.82 |
> > > | **APT (fused)** | **1.38M** | **0.002** | **19,873x** | **3.67** | **0.32** |
> > > | UPT | 1.68M | 0.001 | 44,939x | 1.03 | 4.33 |
> > >
> > > From an ML systems perspective, HOOD is less efficient than other transformer-based neural baselines considered here. It is **52x slower than APT** (383x vs. 19,873x speedup), uses **10x more memory** (38.5 GB vs. 3.7 GB), and yields **20x higher** saturation error ($\delta S_g$ 6.48% vs. 0.32%). This additional overhead stems from the hierarchical graph pipeline, including per-batch FPS hierarchy construction, multi-scale radius-graph construction, and edge feature encoding at each U-Net graph level. By contrast, APT’s fixed-size latent bottleneck avoids these costs and yields a substantially better accuracy-efficiency trade-off. We will add this discussion and the HOOD baseline results to the revised manuscript.

---

### Official Review · Reviewer_LWMg · 2026-03-11

**Soundness:** 2
**Presentation:** 2
**Significance:** 2
**Originality:** 2
**Overall Recommendation:** 3
**Confidence:** 5

**Summary:**

The authors introduce the Adaptive Physics Transformer (APT), a geometry‑, mesh‑, and physics‑agnostic neural operator designed to reduce computational costs. The proposed method is evaluated using two subsurface energy datasets. The local‑to‑global fusion module operating in latent space appears to enable more accurate modeling of localized interactions while still capturing long‑range dependencies.

**Compliance With Llm Reviewing Policy:**

Affirmed.

**Key Questions For Authors:**

- Regarding the statement “positioning it as a robust and scalable backbone for large-scale subsurface foundation model development,” it would be helpful to clarify the justification for this claim. The current experiments use models in the range of O(1M)–O(10M) parameters, which is substantially smaller than what is typically recognized as a foundation-model scale in the community. It would also be useful to more explicitly define what aspect of the work constitutes a “foundation model,” as this is not yet clearly articulated in the manuscript. For instance, how about the fine-tuning performance of this model?
- It would also strengthen the paper to provide evidence related to scaling behavior. If the intention is to position the architecture as scalable, a scaling experiment for either empirical or analytical would help the claim.
- On modern accelerator, a 96.3M‑parameter model generally does not run significantly slower than a 17.7M‑parameter model (Even the 700M model may not significantly slower). The performance difference is often marginal in practical settings or the baseline competitor’s model performed slightly better.  Therefore, it would be helpful to avoid the confusion about why parameter size is emphasized in the discussion.

**Limitations:**

Yes

**Strengths And Weaknesses:**

Strengths:
- APT outperforms community‑standard AI architectures by 10%~ in demonstrated experiments for subsurface simulation tasks, particularly in predicting saturation parameters, which are highly sensitive to nonlinear flow dynamics and boundary‑driven effects. Accurate emulation of saturation behavior within AI models is essential for reliable reservoir forecasting and has impacts on a critical capability in the oil and gas industry.
- The model demonstrates performance on par with state‑of‑the‑art methods despite operating with an order of magnitude fewer parameters.

Weaknesses:
- Novelty claim around AMR and adaptive meshes is overstated. Mesh-native learning on adaptive/unstructured meshes has been extensively studied. For instance, a paper entitled Mesh-Informed Neural Operator : A Transformer Generative Approach proposed Mesh-Informed Neural Operator (MINO) may closely align with the underlying idea of this study.
- While the proposed algorithm demonstrated promising performance based on Table 5, there’s no sign of a fundamentally new operator-learning principle, theory, or unambiguous breakthrough in capabilities beyond existing unstructured-grid methods.
- While the model achieved the compatible performance in x5 times smaller parameter against FNO deep O Net, these parameter numbers are 96M vs 17.9M and there may not show significant advantages.

---

> ### Author Rebuttal · Authors · 2026-03-31
>
> Thank you for the thorough review and helpful suggestions. We address each concern below with clarifications and new experimental evidence.
>
> ## Novelty regarding AMR and comparison with MINO
>
> We appreciate reviewer LWMg raising this concern. We want to clarify that APT addresses a fundamentally different and harder AMR setting than previous mesh-native learning methods. The key distinction is the type of mesh adaptivity.
>
> * **R-adaptive:** moves existing nodes to high-gradient areas without changing element count - mesh remains same topology
> * **H-adaptive:** refines the mesh by increasing or reducing element count - changing mesh topology
> * **HR-adaptive** combines both, optimizing node location and refining size - changing mesh topology
>
> Most previous mesh-native methods, including MINO, MeshGraphNet, transolver, and their variants, operated on R-adaptive or fixed unstructured meshes. They can’t work with HR-adaptive data because they require fixed graph topology across input and output. To our best knowledge, the UPT architecture was the first architecture that can query input and output with different mesh topology, and APT is the first work that demonstrates this capability with actual HR-adaptive (DMO) simulation dataset. We will add this distinction explicitly in the revised manuscript.
>
> Regarding MINO, specifically, MINO was designed for generative task where input and outputs share an irregular but *fixed* mesh. MINO specifically requires input and output at identical node locations, making it incompatible with time-varying DMO meshes. We benchmarked MINO with interpolation preprocessing:
>
> **Table R2. 3D Adaptive Mesh ATES**
>
> | Model           | R² (%)   | Rel. L₂          |
> | --------------- | -------- | ---------------- |
> | UPT             | 96.02    | 0.0401           |
> | Transolver      | 98.60    | 0.0237           |
> | MINO            | —        | OOM on 80GB A100 |
> | **APT (fused)** | **99.0** | **0.011**        |
>
> We also ran new baseline comparisons on the faulted CO₂ benchmark, where MINO can fit in memory:
>
> **Table R1. 2D Faulted CO₂ Storage**
>
> | Model           | δPg (%)  | δSg (%)  |
> | --------------- | -------- | -------- |
> | MINO            | 0.94     | 6.89     |
> | Transolver      | 0.87     | 3.82     |
> | UPT             | 0.43     | 4.33     |
> | MGN-LSTM        | 0.20     | 1.20     |
> | **APT (fused)** | **0.11** | **0.32** |
>
> MINO achieves 6.89% saturation error — 21.5× worse than APT. This confirms that MINO's mesh-informed attention, designed for fixed-topology meshes with relatively smooth fields, does not address the multi-scale heterogeneity challenge that APT's fused local-global encoder is specifically designed to resolve.
>
> ## Foundation model claim and scaling evidence
>
> We will revise to clarify APT is a **scalable backbone architecture** satisfying prerequisites for foundation model development, rather than claiming to be a foundation model at current scale.
>
> **We have now conducted scaling experiments on the ATES benchmark**, varying both data and model size:
>
> **Table R3. Data Scaling (fixed 17.5M parameters)**
>
> | Training Samples | Best Test Loss |
> | ---------------- | -------------- |
> | 40k              | 0.1153         |
> | 120k             | 0.0829         |
> | 161k             | 0.0812         |
>
> **Table R4. Model Scaling (fixed 100% data)**
>
> | Parameters | Best Test Loss |
> | ---------- | -------------- |
> | 0.8M       | 0.1167         |
> | 17.5M      | 0.0812         |
> | 69.4M      | 0.0722         |
>
> Both axes show **monotonic improvement with no saturation**:
>
> * **Data scaling**: 4× more data (40k → 161k) reduces test loss by 29.6% (0.1153 → 0.0812), with continued improvement at 161k samples — indicating the model has not yet saturated on available data.
> * **Model scaling**: Scaling from 0.8M to 69.4M parameters (87×) reduces test loss by 38.1% (0.1167 → 0.0722). This confirms that APT benefits from increased capacity in a manner consistent with scaling laws observed in other domains.
>
> APT demonstrates favorable scaling behavior along **both the data and model size**, supporting its position to serve as a scalable backbone for subsurface use cases.
>
> We also acknowledge that fine-tuning performance is an important aspect of foundation model performance. Given the limited rebuttal time frame, we were not able to conduct a fine-tuning experiment. We will include this in the limitation.
>
> ## Parameter size discussion
>
> We agree that inference latency differences between 17.9M and 96.3M are often marginal on modern accelerators. We will revise to emphasize the more practically significant advantage: 1 unified model vs. 5 separate models. Nested FNO and FNO-DeepONet are structurally locked to a specific mesh hierarchy — changing the LGR configuration (e.g., from 5 levels to 3, as in our Channelized dataset) requires redesigning and retraining the entire pipeline. APT handles arbitrary configurations with the same architecture and a single training run.

---

> > ### Author Rebuttal · Reviewer_LWMg · 2026-04-02
> >
> > Thanks for the time and effort for rebuttal. The authors provided a clear novelty clarification, and additional experiments
> > (e.g., Table R1. 2D Faulted CO₂ Storage), but I believe the absolute performance gains over strong baselines are relatively modest in some settings. Therefore, I decided to maintain my score.

---

### Official Review · Reviewer_L35E · 2026-03-11

**Soundness:** 4
**Presentation:** 3
**Significance:** 4
**Originality:** 3
**Overall Recommendation:** 5
**Confidence:** 3

**Summary:**

The paper develops a model architecture for predicting fields of subsurface energy sytems. The encoder architecture combines an attention-based global encoder and a graph neural operator for local dependencies. The decoder uses cross attention to predict the value at each query location. This way, the method is adaptive to unstructured meshes and does not require strict grids or interpolation. The method is benchmarked on diverse tasks against multiple baselines including fourier neural operators and graph-based methods.

**Compliance With Llm Reviewing Policy:**

Affirmed.

**Final Justification:**

I was positive about this paper from the beginning since it proposed a well-motivated approach for an important problem which achieved superior performance than baselines on several benchmarks. In particular, the benchmark tasks were quite diverse, showing the generality of the proposed approach. The rebuttal helped to clarify some parts I did not fully understand, and it reinforced my prior assessment.

**Key Questions For Authors:**

1) The selection of baseline methods is a bit unclear to me. FNO is used as a baseline for several experiments, but then also some variations appear (U-FNO), and other methods such as U-Net are only used for few tasks. How were the baselines selected? Is it based on SOTA per benchmark, or are some methods (e.g. U-Net) just not applicable to other tasks?
2) It was mentioned that additional cross attention layers improve the performance of the model. Could increasing the model size help to outperform larger models e.g. in Table 4? How does the inference time compare in this case - is the ATP model slower due to the attention operation, despite using smaller models?
3) Section 4.3 shows that cross-section training improves the performance. However, as far as I understand, the model was trained separately for each of the experiments in section 3. Is the reason that the tasks are just too different, or could there be a benefit in cross-task training?

**Limitations:**

yes

**Strengths And Weaknesses:**

Strengths:
(1) The method is motivated well and the model design is appropriate to capture global and local dependencies.
(2) There are extensive experiments showing the superiority of the method in diverse tasks. It is important to note that the benchmarks cover very different tasks, showing the general applicability of the method.
(3) The problem setting is well-defined and understandable.
(4) The study on super resolution capability is particularly interesting considering recent work showing that other methods do not fulfill this promise.

Weaknesses:
(1) The novelty is limited since it's a combination of well-known architectures. It seems to be mainly an engineering effort leading to the performance gains.
(2) Other methods (FNO-DeepONet) are better than the proposed method on one benchmark dataset; however, those models are significantly larger.

---

> ### Author Rebuttal · Authors · 2026-03-31
>
> Thank you for the thorough review and the helpful suggestions!
>
> **Q1.** The baseline models were selected mainly based on published SOTA per benchmark.
>
> * **CO2 storage for Irregular Grid with Faults — MGN and MGN-LSTM:** Ju, Xin, et al. *"Learning CO2 plume migration in faulted reservoirs with Graph Neural Networks."* Computers & Geosciences 193 (2024): 105711.
> * **Hydrocarbon in Cartesian Grid — FNO and U-FNO:** Badawi, Daniel, and Eduardo Gildin *"Neural operator-based proxy for reservoir simulations considering varying well settings, locations, and permeability fields."* Computers & Geosciences 196 (2025): 105826.
> * **Basin-Scale CO2 Storage with LGR — Nested FNO:** Wen, Gege, et al. *"Real-time high-resolution CO 2 geological storage prediction using nested Fourier neural operators."* Energy & Environmental Science 16.4 (2023): 1732-1741.
> * **FNO-DeepONet:** Lee, Jonathan E., et al. *"Efficient and generalizable nested Fourier-DeepONet for three-dimensional geological carbon sequestration."* Engineering Applications of Computational Fluid Mechanics 18.1 (2024): 2435457.
>
> For the **Adaptive Mesh - Aquifer Thermal Energy Storage** dataset, no prior SOTA benchmark is available so we run benchmarks with U-Net and FNO.
>
> In response to reviewer feedback, we have also added **MINO, Transolver, and UPT** as additional baselines across multiple benchmarks:
>
> | Model      | Faulted CO₂ δSg (%) |              ATES Rel. L₂ |
> | ---------- | ------------------: | ------------------------: |
> | MINO       |                6.89 | — (cannot run due to OOM) |
> | Transolver |                3.82 |                    0.0237 |
> | UPT        |                4.33 |                    0.0401 |
> | APT        |                0.32 |                     0.011 |
>
> We also evaluated on three general PDE benchmarks (**Airfoil, Darcy, Elasticity**), where APT is consistently the second-best model behind Transolver. This confirms that APT's advantage is most pronounced on heterogeneous, multi-scale systems (where it outperforms Transolver by 11.9×), while remaining competitive on homogeneous-domain problems.
>
> ---
>
> **Q2.** Can scaling APT close the gap in Table 4? We have now conducted model scaling experiments on the ATES benchmark:
>
> | Model | Params | Dim | Inference Time (ms) | Mem (MB) | R² (%) | Val Loss |
> | ----- | -----: | --: | ------------------: | -------: | -----: | -------: |
> | Small |  0.79M |  48 |                6.46 |       80 |  98.49 |   0.1064 |
> | Base  | 17.47M | 192 |                8.27 |      268 |  99.26 |   0.0734 |
> | Large | 69.37M | 384 |               10.68 |      636 |  99.39 |   0.0722 |
>
> Performance improves monotonically with parameter count, suggesting that scaling APT further could potentially close the remaining saturation accuracy gap against Nested FNO in Table 4.
>
> **Inference time vs. mesh size.** We also profiled APT's runtime as a function of input cell count on the basin-scale LGR benchmark:
>
> | Number of Cells | Graph Pooling (%) |
> | --------------- | ----------------: |
> | 300k            |              ~47% |
> | 550k            |              ~60% |
> | 800k            |              ~64% |
> | 1.0M            |              ~70% |
>
> Total inference time scales roughly linearly with cell count. Notably, graph radius pooling (the local GNO branch) becomes the dominant cost at larger meshes, accounting for ~70% of runtime at 1M cells. The latent-space attention operations (approximator and decoder) contribute a roughly constant overhead regardless of mesh size.
>
> ---
>
> **Q3.** Cross-dataset training is only tested for section 4.3 where the two datasets were simulations of the same physical process (i.e., CO2 migration in saline water) with different input distributions and resolutions. Under this set up, we can directly train APT without any architectural modification.
>
> Cross-task training is a natural and exciting future extension of this work. When using APT for cross-task training, additional development is needed to handle different physical quantities of interest across tasks. Methods to incorporate different physics quantities is an open research question which we will explore in future work, building on top of the mesh-agnostic capability that APT provides.
>
> We thank the reviewer again for the supportive evaluation and insightful questions. We will incorporate the additional baselines, scaling results, and inference time analysis in the revision.

---

> > ### Author Rebuttal · Reviewer_L35E · 2026-04-02
> >
> > Thanks for the clarifications, my questions were all addressed. I will keep my positive rating.

---

### Decision · Program_Chairs · 2026-04-30

**Decision:**

Accept (regular)

**Comment:**

This paper (APT) fuses local graph encoding and global attention for accurate surrogate modeling on heterogeneous subsurface meshes.
Reviewers value the important problem, broad experiments, and effective local-global design; the rebuttal clarified baselines, scaling, efficiency, and domain-specific advantages.
Main concerns were incremental novelty, initially incomplete baseline coverage, overstated foundation-model/physics-agnostic framing, variance reporting, and one LGR benchmark where specialized hierarchical baselines still performed better on saturation.

Post rebuttal: strong, broadly validated contribution with remaining novelty/framing issues substantially addressed. I would suggest an acceptance (scores 5/5/4/3)